# Chemical control of colloidal self-assembly driven by the electrosolvation force

Sida Wang [1,3], Rowan Walker-Gibbons[1,3], Bethany Watkins [1], Binghui Lin[1] & Madhavi Krishnan [1,2] ✉

Self-assembly of matter in solution generally relies on attractive interactions that overcome entropy and drive the formation of higher-order molecular and particulate structures. Such interactions are central to a variety of molecular processes, e.g., crystallisation, biomolecular folding and condensation, pathological protein aggregation and biofouling. The electrosolvation force introduces a distinct conceptual paradigm to the existing palette of interactions that govern the spontaneous accretion and organisation of matter. However, an understanding of the underlying physical chemistry, and therefore the ability to exert control over and tune the interaction, remains incomplete. Here we provide further evidence that this force arises from the structure of the interfacial electrolyte. Neutral molecules such as a different solvent, osmolytes or surfactants, may − even at very low concentrations in the medium − disrupt or reinforce pre-existing interfacial solvent structure, thereby delivering unanticipated chemical tuning of the ability of matter to self-assemble. The observations present unexpected mechanistic elements that may explain the impact of co-solvents and osmolytes on protein structure, stability and biomolecular condensation. Our findings thus furnish insight into the microscopic mechanisms that drive the emergence of order and structure from molecular to macroscopic scales in the solution phase.

Liquids in contact with interfaces play a pivotal role in a range of natural phenomena occurring, e.g., in the atmosphere, in geology, chemistry and biology[1–3]. The solid-liquid interface, in particular, plays a crucial role in a host of scientific and technological areas such as chromatography, electrochemical energy generation and storage, biological implant design, biofouling, self-assembly, drug delivery and the stability of pharmaceuticals and fine chemicals[4–8]. Molecular level phenomena at an interface with an electrolyte can impact macroscopic system properties and behaviour. For instance, interfacial molecular processes underpin the uptake of gases at the ocean/air interface, and the adsorption of surfactants to interfaces is central to a range of practical and therapeutic applications[9]. Furthermore, osmolytes are small, net-neutral soluble compounds that are known to affect the folding, stability, aggregation, and condensation of biomolecules,

enzymatic activity, and surface tension[10–15]. Knowledge of how interfacial processes affect macroscopic system behaviour is therefore central to our ability to understand, design and engineer system properties in phenomena that span a range of length scales in a variety of areas.

Here, we provide molecular-level insight into the recently described electrosolvation force, demonstrating routes to chemically tuning and controlling long-ranged interactions that govern accretion, structure formation and self-assembly in suspended matter[16–18]. Specifically, we provide evidence that the electrosolvation force is driven by a significant normal component of molecular dipole moments at an interface. We further suggest that chemical modulation of this interaction via small molecules in the medium may occur by both disruptive and reinforcing effects on the interfacial solvent structure.

[1]Physical and Theoretical Chemistry Laboratory, Department of Chemistry, University of Oxford, Oxford, UK. [2]The Kavli Institute for Nanoscience Discovery, Oxford, UK. [3]These authors contributed equally: Sida Wang, Rowan Walker-Gibbons. ✉e-mail: madhavi.krishnan@chem.ox.ac.uk

The electrosolvation force refers to an electrostatically mediated solvent- or solvation-dependent interparticle interaction that arises from molecular orientational structuring in the electrolyte in the vicinity of a charged surface in contact with a fluid phase[16–19]. It has been previously demonstrated that like-charged particles in solution may attract or repel depending on the nature of the solvent and the sign of charge of the particle. This counterintuitive long-range attraction, therefore, effectively breaks the symmetry associated with inversion in the sign of charge in the system, which is characteristic of the Coulombic interaction[18,20,21]. Thus, in aqueous solution, negatively charged colloidal particles were observed to attract at long range (5–10 Debye lengths, $\kappa^{-1}$) while positively charged particles repelled. Conversely, positively charged particles suspended in short-chain alcohols were found to attract, whereas negatively charged particles repelled[18]. Here, the Debye length $\kappa^{-1} = \sqrt{\epsilon_0 \epsilon_r k_B T/(2\rho_{ion} e^2)}$ represents a length scale over which the electrical potential due to a charged object decays exponentially with distance in an electrolyte containing monovalent ions at a number density $\rho_{ion}$ in a medium of dielectric constant $\epsilon_r$. Further, $\epsilon_0$ is the permittivity of free space, $e$ is the elementary charge, $k_B$ is the Boltzmann constant and $T$ is the absolute temperature.

Within the 'interfacial solvation model', the effective interparticle interaction potential, which has been corroborated in experiments, may be written as

$$U_{tot} = \Delta F_{el} + \Delta F_{int} \approx A \exp(-\kappa_1 x) + B \exp(-\kappa_2 x) \quad (1)$$

The first term, $\Delta F_{el} > 0$, represents the familiar repulsive electrostatic interaction free energy between two large like-charged spheres of radius $R$ and intersurface separation $x$, in the regime $R \gg \kappa^{-1} = \kappa_1^{-1}$. The second term, $\Delta F_{int} \approx B \exp(-\kappa_2 x)$, represents the interfacially generated free-energy contribution to the interaction[16,17]. The model also suggests that $\kappa_2 < \kappa_1 \approx \kappa$, and it can be shown that

$$B \propto z\,\mu_{av}(\sigma) \quad (2)$$

where $z$ is the sign of the charge of the ionised groups carried by a particle of surface charge density $\sigma$. Importantly, in our model, $\mu_{av}(\sigma) \approx \mu_{av}(\sigma = 0)$, which holds for low magnitudes of $\sigma$, is the excess normal dipole moment surface density at the interface between the object and the electrolyte, and can be obtained from molecular dynamics (MD) simulations. Importantly, a non-zero average dipole moment density at an interface ($\mu_{av} \neq 0$) implies an additional, previously overlooked contribution to the interparticle interaction energy, as indicated in Eqs. (1) and (2). Note that at interfaces in pure solvents, a net orientation of solvent molecules entailing $\mu_{av} \neq 0$ has been reported on extensively in interfacial spectroscopy experiments, especially for the silica-water interface[22–24]. In previous studies, we formulated the interfacial free energy contribution in Eqs. (1) and (2) in terms of an interfacial solvation-goverend electrical potential, $\varphi_{int}$[16–19]. Since $\mu_{av}(\sigma) = -\epsilon_0 \varphi_{int}(\sigma)$, we phrase the discussion in this work in terms of the more intuitively accessible interfacial dipole moment density $\mu_{av}$ (MD simulation methods, Supplementary Note. 1). Further, Eq. (1) excludes any contribution from the van der Waals (vdW) force which is negligible at the interparticle separations relevant to these experiments (BD simulation methods).

Thus, depending on the sign and magnitude of $B$ in the electrosolvation or interfacial term in Eq. (2), the interaction between electrically like charged particles in solution may either be net-attractive ($B < 0$) or repulsive ($B \gtrsim 0$) at long range, and in the former case, can result in a long-ranged minimum of depth $w$ in the pair-interaction potential (Fig. 1f). Specifically, on a plot of $\mu_{av}$ vs. $\sigma$, the model would support the appearance of interparticle attraction in the quadrants reflecting opposite signs of the two quantities (coloured regions in Fig. 2e). Pair potentials entailing minima of tunable depth play a crucial role in fostering tailored assembly of large-scale collections of particles[18]. In this work, we explore the ability to chemically tune the value of $B$ in particle interactions based on our understanding of the molecular mechanisms underpinning the electrosolvation force. The study concludes with a comparison of

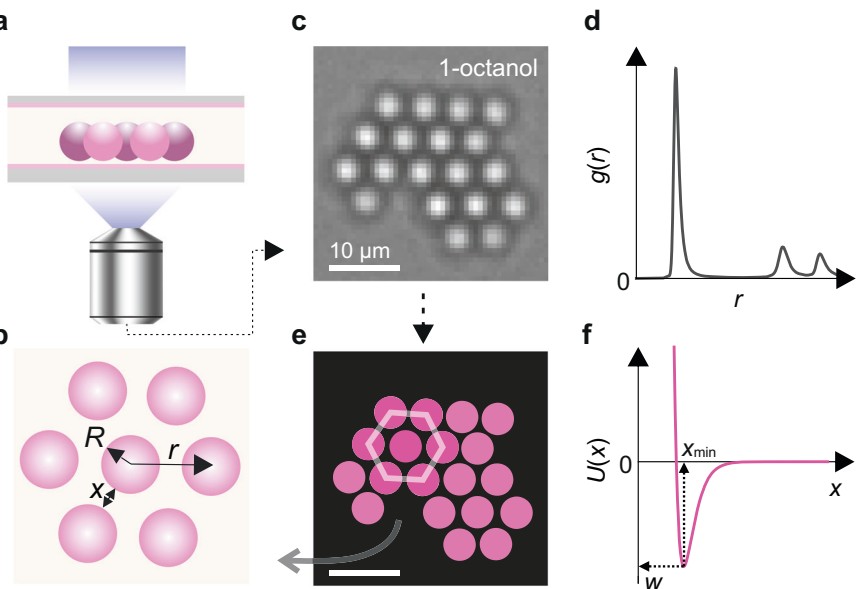

**Fig. 1 | Experimental set-up for inferring interparticle interactions in 2-d colloidal suspensions. a, b** Schematic representation of a cluster of spheres of radius in a gravity-sedimented colloidal suspension, levitating above a silica coverglass, at an interparticle separation $r$ and intersurface separation $x$. Particles are imaged using bright field microscopy (see Experimental methods). **c, d** Particle coordinates are extracted from a microscope image using single particle tracking software[65] and used to generate radial distribution profiles, $g(r)$. **e** Digitised microscopy image where particles are represented as coloured discs of uniform diameter $2R$ on a black background. Like-charged colloidal particles can be observed to form stable, slowly reorganising hexagonally close-packed (hcp) clusters in solution. Scale bar 10 μm. **f** Brownian dynamics simulations are used to infer a pair-interaction potential, $U(x)$, characterised by an attractive minimum of depth $w$, and location $x_{min}$, capable of reproducing the experimentally observed $g(r)$ (see BD Simulation methods). Inferred interaction strengths $|w| \approx 5\,k_B T$ are typical for strongly clustering systems, as shown here for positively charged $NH_2$ particles suspended in 1-octanol.

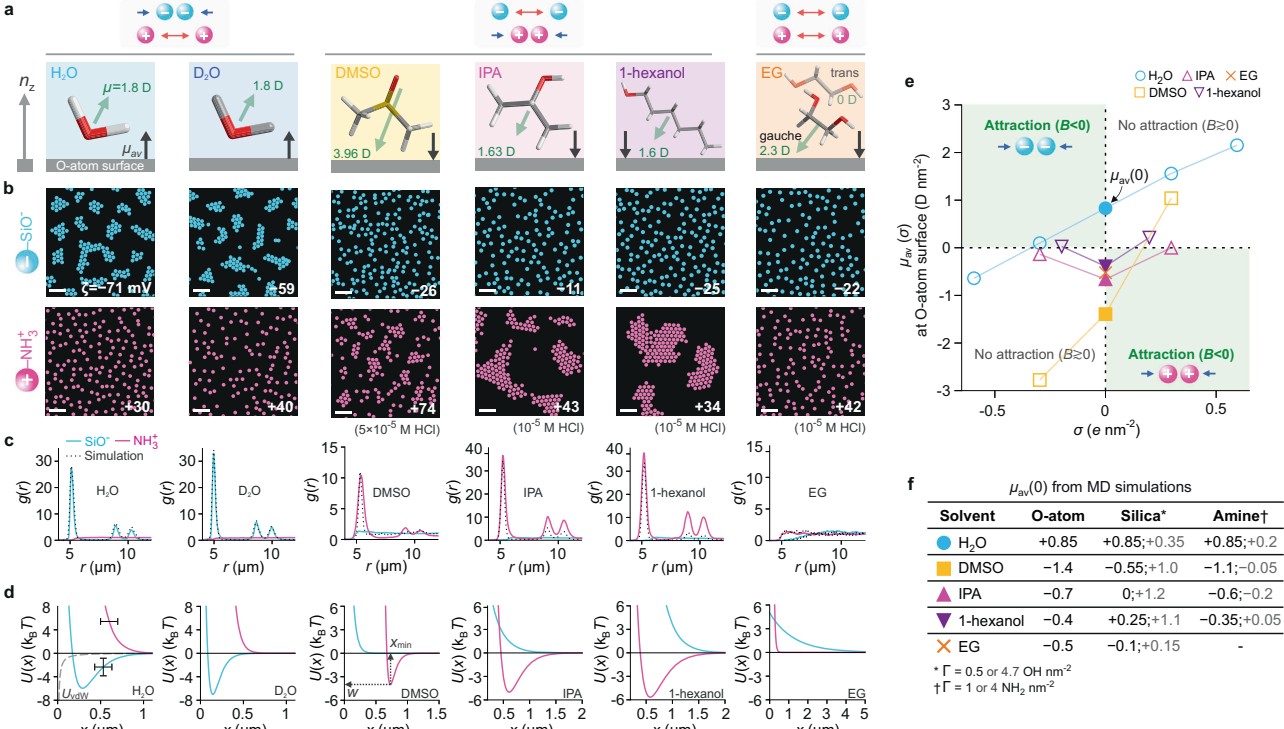

**Fig. 2 | Solvent dependence of charge-asymmetric cluster formation.**
**a** Structures and average orientation of solvent molecules at an uncharged O-atom surface (dipole moments − green arrows; surface normal − $n_z$), as inferred from MD simulations. The normal component of the net interfacial molecular dipole moment, $\mu_{av}$, points slightly away from the surface for $H_2O$ and $D_2O$ and towards the surface for the other cases. **b** Digitised experimental images of negatively charged silica $SiO_2$ particles (blue, $2R = 4.82$ or $5.17$ μm) and positively charged aminated silica $NH_2$-$SiO_2$ particles (pink, $2R = 4.63$ μm) suspended in $H_2O$, $D_2O$, DMSO, 2-propanol (IPA), 1-hexanol, and ethylene glycol (EG) (Supplementary Fig. 11). Scale bars 20 μm. Stable clusters were observed in $H_2O$ and $D_2O$ for $SiO_2$ particles, and in DMSO, IPA and in primary and secondary alcohols for $NH_2$ particles (Fig. 3), but not in EG (Supplementary Fig. 17, Supplementary Note 3). Measured zeta potentials, $\zeta$ (in mV) provided in inset. **c** Experimental $g(r)$ for $SiO_2$ (solid blue lines) and $NH_2$ particles (solid pink lines), and corresponding simulated $g(r)$ profiles (dashed grey lines). Ordered clusters yield $g(r)$ profiles with a periodic peak

structure indicating interparticle attraction. **d** Pair-interaction potentials $U(x)$ of the form of Eq. (1), inferred from BD simulations, that best reflect the experimental $g(r)$ profiles for $SiO_2$ (blue) and $NH_2$ (pink) particles, and the negligible vdW contribution, $U_{vdW}$ (grey dashed line) (Supplementary Table 25). Error bars denote estimated uncertainties of $\pm 100$ nm on particle diameter and $\pm 1.5$ $k_B T$ in $w$.
**e** Excess interfacial dipole moment density $\mu_{av}(\sigma)$ for various solvents at surfaces carrying positive and negative surface charge density $\sigma$, as obtained from MD simulations. At zero surface charge, $\sigma = 0$, we obtain $\mu_{av}(0) = +0.85$ D nm$^{-2}$ for water and $\mu_{av}(0) < 0$ for other solvents (filled symbols). Parameter values in green quadrants of the $\mu_{av}$ vs. $\sigma$ plot support interparticle attraction and cluster formation between electrically like-charged particles (since $B < 0$ in Eqs. (1) and (2)).
**f** $\mu_{av}(0)$ for solvents in contact with a neutral O-atom surface, silica surfaces with varying surface group density ($\Gamma = 0.5$ or $4.7$ OH nm$^{-2}$), and model amine surfaces with varying surface group density ($\Gamma = 1$ or $4$ $NH_2$ nm$^{-2}$).

## Results and discussion

the experimental observations with the indications of the interfacial solvation model.

We report on a range of experiments examining the electrosolvation interaction between charged particles of different surface chemistry and charge, suspended in aprotic and protic solvents, in solvent mixtures, as well as in aqueous electrolytes containing varying amounts of co-solvents such as zwitterionic osmolytes and surfactants. We observed the spatial structure of two-dimensional gravity-sedimented suspensions of colloidal particles using bright field microscopy, as described previously[18] (Fig. 1). We worked with two different samples of silica ($SiO_2$, diameter $2R = 4.82$ or $5.17$ μm) and carboxyl-functionalized melamine resin (COOH-MF or COOH; $2R = 5.29$ μm) representing negatively charged microspheres, and positively charged aminated silica ($NH_2$-$SiO_2$ or $NH_2$; $2R = 4.63$ μm) microspheres (see experimental methods). Measured radial probability density distributions, $g(r)$, were compared with the results of Brownian dynamics (BD) simulations of a 2-d system of particles to infer the underlying pair-interaction potential, $U(x)$, as previously described (BD simulations methods)[18]. Furthermore, MD simulations of the respective particle surface types in contact with the fluid phase

of interest provide microscopic insight into the properties of the interfacial electrolyte, i.e., molecular densities and orientations, as well as estimates of the total dipole moment density, $\mu_{av}$, and of $\mu_{av,w}$, the dipole moment density due to water alone, under various experimental conditions.

We expect the various experimental conditions tested to alter the interfacial electrolyte structure, influencing the value of $\mu_{av}$ or $\mu_{av,w}$ compared to the pure electrolyte in different ways. Such experiments provide a means to put a core feature of the interfacial solvation model to a systematic test. Values of the surface-averaged excess normal dipole moment surface density inferred from MD simulations provide a theoretically expected sign and approximate magnitude of $B$ in Eq. (2) under various conditions. Experiments reveal either the presence or absence of interparticle attraction, and these outcomes may be compared with qualitative indications of the model (MD simulation methods, Supplementary Note 1)[16–18]. The study concludes with a diagrammatic summary of experimental observations, i.e., the presence or absence of attraction, as a function of system properties $\sigma$ and $\mu_{av}$. Owing to the significant challenges associated with experimentally determining or estimating $\mu_{av}$ at the surface of a microsphere, we use values inferred from molecular simulations in each case. The experimental findings largely agree with expectations from the model,

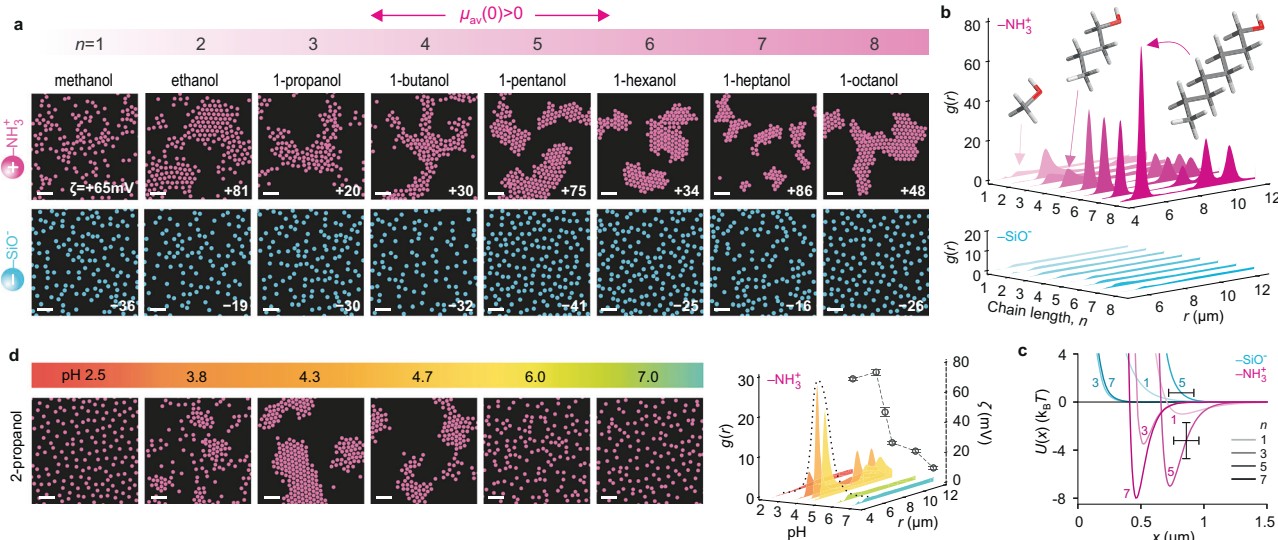

**Fig. 3 | Cluster formation in suspensions of positively charged particles in primary alcohols. a** Representative image of positively charged aminated silica $NH_2$-$SiO_2$ particles (pink) and negatively charged $SiO_2$ particles (blue) in primary alcohols of increasing chain length from methanol ($n=1$, left) to 1-octanol ($n=8$, right) (see Supplementary Table 4 for experimental conditions). All amine particle experiments contained $10^{-5}$ M HCl. Measured zeta potentials ($\zeta$) are noted. **b** Radial probability density distributions as a function of alcohol chain length for the experiments shown in (**a**). **c** Inferred pair-interaction potentials, $U(x)$, from BD simulations that best reproduce the experimental $g(r)$s for select cases (see Supplementary Table 26 for parameters). Well depths $|w| \approx 1\,k_BT$ and $\approx 3\,k_BT$ were

sufficient to capture the $g(r)$ profile of $NH_2$ particles in methanol and 1-propanol, respectively. Larger values $|w| \approx 7\,k_BT$ were required to capture the strong cluster formation observed in the higher chain alcohols 1-pentanol and 1-heptanol. Purely repulsive pair potentials described interactions of negatively charged $SiO_2$ particles in alcohols (Supplementary Table 26). Error bars denote estimated uncertainties of $\pm 100$ nm on particle diameter and $\pm 1.5\,k_BT$ in $w$. **d** Effect of pH on cluster formation, shown for $NH_2$ particles in 2-propanol (IPA). Negatively charged $SiO_2$ and COOH particles in IPA display no evidence of interparticle attraction as a function of pH (Supplementary Fig. 5). Scale bars 20 μm.

lending support to the view that interfacial solvent structuring can play a central role in long-range forces between objects in solution.

## The electrosolvation force is observed in many solvents

The solvent itself provides the most direct means of probing the microscopic origins of a putative solvation-governed long-ranged force between suspended particles. We therefore examined interparticle interactions in a range of protic and aprotic solvents namely, water, heavy water, dimethyl sulfoxide (DMSO), methanol, ethanol, 1-propanol, 2-propanol (IPA), 1-butanol, 1-pentanol, 1-hexanol, 1-heptanol, 1-octanol and ethylene glycol (EG) (Figs. 2, 3 and Supplementary Notes 2–4).

Heavy water, $D_2O$, presents the closest chemical analogue to light water, $H_2O$ – the solvent in which a strong signature of the electrosolvation force was first identified[18]. Possessing highly similar electronic properties to water (such as dipole moment dielectric constant), $D_2O$ molecules predictably exhibit orientational behaviour at interfaces similar to $H_2O$, as reported both in MD simulations and in experiments[24–26]. Assuming a value of $\mu_{av}(\sigma = 0) = \mu_{av}(0) \approx +0.8$ D $nm^{-2}$, similar to that of water, would suggest $B < 0$ for negatively charged particles ($z = -1$) in Eq. (2). Within the electrosolvation view, this would imply interparticle attraction and hence the possibility of the formation of stable clusters. Indeed, experiments on silica particles in $D_2O$ displayed the formation of stable, slowly reorganising hexagonally closed packed (hcp) clusters, characterised by an average particle inter-surface separation $x_{min} \approx 2.5\kappa^{-1}$ which is in line with indications from the interfacial solvation model[16–18] (Supplementary Tables 3, 25). On the other hand, for positively charged particles where $z = +1$, a value of $\mu_{av}(0) > 0$ entails $B > 0$ which in turn implies interparticle repulsions and no cluster formation, capturing the experimental observation for $NH_2$ particles suspended in $D_2O$ (Fig. 2 and Supplementary Fig. 18).

We then explored interparticle interactions in DMSO, a polar aprotic solvent whose trigonal molecular structure echoes the bent

structure of water. Interestingly, in contrast to water, we found that positively charged aminated silica particles formed clusters in DMSO, albeit in a narrow range of pH, controlled experimentally by varying the amount of HCl added to the pure solvent (Fig. 1, Supplementary Fig. 16, and Supplementary Table 8). On the other hand, silica particle interactions remained repulsive over a range of added HCl and NaOH concentrations (Supplementary Fig. 16). The observations on $NH_2$ particles are in fact, consistent with the expected orientation of DMSO from MD simulations performed at an O-atom or model $NH_2$ surface (Fig. 2f)[19]. Here the net orientation of solvent dipoles is inverted compared to water, i.e., $\mu_{av}(\sigma) < 0$, for surfaces with a low positive surface charge density, implying interparticle attraction in the interfacial solvation model (Fig. 2e). Simulations at model silica surfaces, however, indicate that $\mu_{av}(0)$ depends on the silanol group density, $\Gamma$, and may be negative or positive depending on the value of $\Gamma$, which is similar to inferences for alcohols[19] (Fig. 2f). But for silica surfaces with $\Gamma \lesssim 1\,nm^{-2}$ – which may be a realistic value – MD simulations indeed suggest that $\mu_{av}(0) \lesssim 0$ implying $B \gtrsim 0$. This points to an interparticle interaction that is devoid of a significant attractive contribution for negatively charged silica particles, as observed in the experiment (Supplementary Notes 4, 8)[27–29].

Next, we examined interparticle interactions in a range of primary alcohols (Fig. 3). Previous experiments showed that positively charged $NH_2$ particles readily self-attract and form crystalline clusters in ethanol and IPA, characterised by large interparticle spacings ($x_{min} \approx 0.5 - 1.5$ μm $= 5 - 11\kappa^{-1}$)[18]. In addition to their aliphatic groups of variable length, primary alcohols from methanol to 1-octanol possess a hydroxyl group capable of hydrogen bonding. Similar to previously described MD simulations of ethanol and IPA at an O-atom interface, we expect the likely net orientation of alcohols at a neutral aminated silica surface to entail the hydroxyl group pointing on average more towards the bulk medium, since this orientation presumably favours hydrogen bonding[18,19]. We estimate a net dipole moment surface density $\mu_{av}(0) = -0.4$ D $nm^{-2}$ for simulations of

1-hexanol at a neutral O-atom wall, similar to both the value expected for other primary alcohols and IPA, as well as results for 1-hexanol and IPA at an amine surface (Fig. 2f). Equation (2) therefore implies interparticle attraction for NH$_2$ particles and repulsion for negatively charged particles, which was indeed observed in experiments performed in a range of primary alcohols (Fig. 3). The strength of the interparticle interaction in our experiments is characterised by the depth, $w$, of the minimum in the inferred pair potential, $U(x)$. Interestingly, similar to cluster formation in water, we found that even in non-aqueous solvents, $w$ depended strongly on the proton concentration in the medium (Fig. 3, Supplementary Fig. 5 and Supplementary Note 2). Note that, similar to DMSO, MD simulations of alcohols at silica surfaces indicate $\mu_{av}(0)$ values that depend strongly on the silanol group density, $\Gamma$, and can point to the possibility of weak cluster formation for silica particles in 1-hexanol, contrary to our experimental observations (Fig. 2f and Supplementary Notes 2, 8).

Given the strong signature of molecular orientational anisotropy of DMSO and primary alcohols at an interface and the apparent impact thereof on interparticle interactions, we set out to examine interparticle interactions in a symmetrical diol, ethylene glycol (1,2-ethanediol − EG). EG has significant conformational complexity compared to water and alcohols: it exists in bulk solution as an 80/20 mix of gauche and trans conformers, where the former conformer has a rather large dipole moment (2.3 D) and the latter an effectively zero dipole moment[30,31]. However, in EG, neither negatively charged SiO$_2$ nor positively charged NH$_2$ particles displayed interparticle attraction under either acidic or basic conditions (Fig. 2 and Supplementary Fig. 17). Interestingly, MD simulations of EG at strongly H-bonding silanol surfaces of group densities $\Gamma = 0.5 − 4.7$ OH nm$^{-2}$ show that $\mu_{av}(0) \approx −0.1$ to $+0.15$ D nm$^{-2}$, which is significantly smaller in magnitude than the value $\mu_{av}(0) \approx −0.5$ D nm$^{-2}$ obtained for EG at a hydrophobic O-atom surface (Fig. 2f, MD simulation methods). It is possible that a low magnitude of $\left|\mu_{av}(0)\right| \approx 0.1$ D nm$^{-2}$ is not large enough to support a substantial attractive force required for cluster formation, i.e., $\mu_{av}(\sigma) \to 0$ would entail $B \to 0$ in Eq. (1) (Supplementary Note 3). EG thus presents an important departure from the other solvent media considered and, therefore, a significant opportunity for further understanding of the electrosolvation interaction.

## Water in alcohol influences the properties of the force

Having examined interparticle interactions in pure solvents, we investigated the nature of the electrosolvation force for positively and negatively charged particles suspended in binary mixtures of water and IPA. NH$_2$ particles formed stable clusters in pure IPA (pH $\approx 4 − 5$) as previously reported, and the addition of small amounts of water up to about 10% (v/v) did not affect the ability of the particles to form clusters (Fig. 4)[18]. For increasing amounts of added water, however, we found that cluster formation was progressively suppressed, with the qualitative structure of the suspensions resembling the results obtained for NH$_2$ particles in pure water (monotonic repulsions).

Negatively charged particles in water-IPA mixtures displayed inverted trends in cluster formation with increasing water content in the mixture (Fig. 4). We observed no clustering in IPA containing small amounts (<1% (v/v)) of water. With increasing fractional content of water, however, silica particles formed stable clusters whose structure above 5% (v/v) water qualitatively reflected that observed in pure water (Fig. 4b). The concentration at which the transition from alcohol-like (repulsive) interparticle interactions to water-like (attractive) interactions was found to depend strongly on particle chemistry and was ≈50% for COOH particles (Fig. 4e).

These intriguing observations do in fact find an explanation within the interfacial solvation view, which would suggest that the sign of $B$ in Eq. (2) is governed by the sign of the dominant interfacial dipole moment characterising a given solvent mixture. In water-alcohol mixtures containing bulk concentrations of ≈10% (v/v) water and higher, the interfacial dipole moment density at the hydrophilic surfaces characteristic of our experiments may be expected to be largely determined by the interfacial character of molecular water[32]. MD simulations of 2% (v/v) water/IPA mixtures indeed showed strong adsorption of water molecules to surface sites bearing silanol groups, indicating an effective concentration enhancement of about a factor 8 compared to the bulk. In contrast, simulations at the same low concentration reveal no substantial surface adsorption of water at a model amine surface which may be viewed as less hydrophilic compared to silica (Fig. 4f, MD simulation methods and Supplementary Note 1). Furthermore, we observed particle-type dependent disparities in the fractional water content marking the onset of cluster formation in negatively charged polymeric COOH and SiO$_2$ particles, which may be attributed to the impact of surface chemistry on the composition and structuring of the interfacial electrolyte. Note that estimating molecular orientations and dipole moment densities from simulations of silica and amine interfaces immersed in solvent mixtures is associated with significant challenges (see MD simulation methods). Therefore we do not infer $\mu_{av}(0)$ values from MD simulations and limit the discussion to a qualitative interpretation of the observed behaviour based on the dominant interfacial solvent species indicated by the simulations.

We also examined a possible role for various monovalent and divalent cations in the interaction between silica particles in water and found no significant influence on the interaction at the low ionic strength (ca. 0.1 mM) tested (Fig. 5).

## Chemical gating of the attraction by charge-neutral species

Inspired by the dramatic impact of small amounts of water on the interparticle interaction in binary solvent mixtures, we sought to investigate the influence of net-neutral molecular species such as zwitterionic osmolytes and surfactants added to the medium. Zwitterions and osmolytes are small net-neutral organic molecules, known to play important roles in determining the folded state of proteins, exerting 'protective' and 'disruptive' effects on protein structural stability[33–35]. Zwitterionic surface coatings are also known to display strong antifouling properties[36]. While the precise detail underpinning these effects is not entirely clear, universal mechanisms involving osmolyte influence on water structure have attracted substantial attention[12,35,37]. Indeed optical spectroscopy experiments on interfaces immersed in electrolytes containing zwitterions such as amino acids and osmolytes have reported both interfacial adsorption of these species as well as alterations to the interfacial water structure[38,39] (Supplementary Notes 5, 6). We therefore examined particle interactions in aqueous electrolytes containing the aromatic L-amino acids tyrosine, tryptophan and phenylalanine, leucine − an aliphatic amino acid, proline, and the polar side-chain amino acids: glutamine, serine and glycine. Silica particles were suspended in aqueous solutions containing amino acids at various bulk concentrations, $c_b$, up to 1 M for polar amino acids such as glycine. pH and conductivity were measured for all solutions and remained largely unaffected by the addition of amino acids (Supplementary Tables 14−22), confirming their zwitterionic state in the solution.

We found that amino acids added to the aqueous phase inhibited the formation of particle clusters at low concentrations ($c_b \approx 10^{-4}$ M) representing ca. 10 ppm in relation to the surrounding water. For example, whilst extremely low tyrosine concentrations of $c_b \approx 10^{-5}$ M had no impact on cluster formation, a significant reduction in the strength of the interparticle attraction was observed at around 0.1 mM (Fig. 6b, c). Concentrations of $c_b \approx 0.5$ mM and above suppressed cluster formation entirely, restoring the canonically expected interparticle repulsion between like-charged particles in an aqueous solution. The inferred pair-interaction potentials displayed systematically decreasing depths of minima, $w$, with $w \to 0$ at $c_b > 0.1$ mM for tyrosine (Fig. 6b–d). For the polar amino acids, glycine, serine, and aliphatic proline, however, the transition from attractive to repulsive

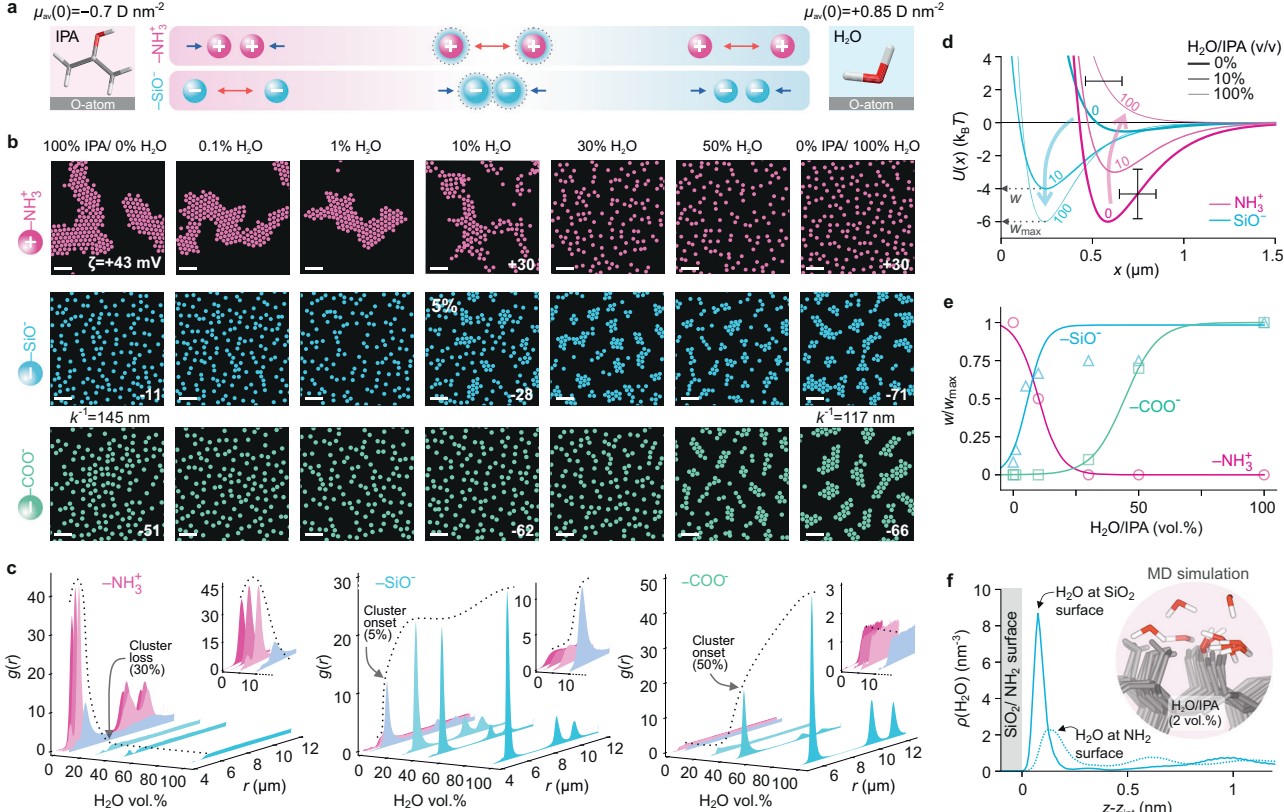

**Fig. 4 | Tuning the strength of the interparticle attraction in water-alcohol mixtures. a** Structure of colloidal particle suspensions for positively charged $NH_2$ (pink), and negatively charged $SiO_2$ (blue) and COOH particles (green), in water/IPA mixtures of increasing volume percentage of water (left to right). **b** Measured zeta potentials of the particles ($\zeta$) are shown in inset. $NH_2$ particles display attraction and cluster formation in pure IPA containing water at a concentration $\lesssim$10% (v/v). $SiO_2$ and COOH particle interactions are purely repulsive in pure IPA, but the onset of interparticle attraction and cluster formation occurs at around 5 and 50% (v/v) water, respectively (Supplementary Table 7 for experimental conditions). Scale bars 20 μm. **c** $g(r)$ profiles as a function of water volume percent for the three particle surface types. **d** Inferred pair-interaction potentials, $U(x)$, from BD simulations that match the experimental particle distributions (see Supplementary

Table 27 for parameters). Error bars denote estimated uncertainties of ± 100 nm on particle diameter and ±1.5 $k_B T$ in $w$. **e** Normalised pair-interaction potential well depths, $w/w_{max}$, as a function of water volume percent. **f** Profiles of interfacial water density, $\rho(H_2O)$ as a function of distance $z$ from the interface, situated at $z_{int}$, calculated from MD simulations of a 2% (v/v) water/IPA mixture in contact with a silica (solid line) and model amine surface (dashed line) carrying surface group densities $\Gamma \approx 4.7$ and 4 $nm^{-2}$ respectively (see MD simulation methods). Simulations indicate significant water adsorption at the hydrophilic silica surface, in contrast with a model surface composed of amine groups. Inset: molecular dynamics simulation snapshot, with interfacial water molecules in a background medium of IPA (pink) adsorbed to silanol groups on the silica surface.

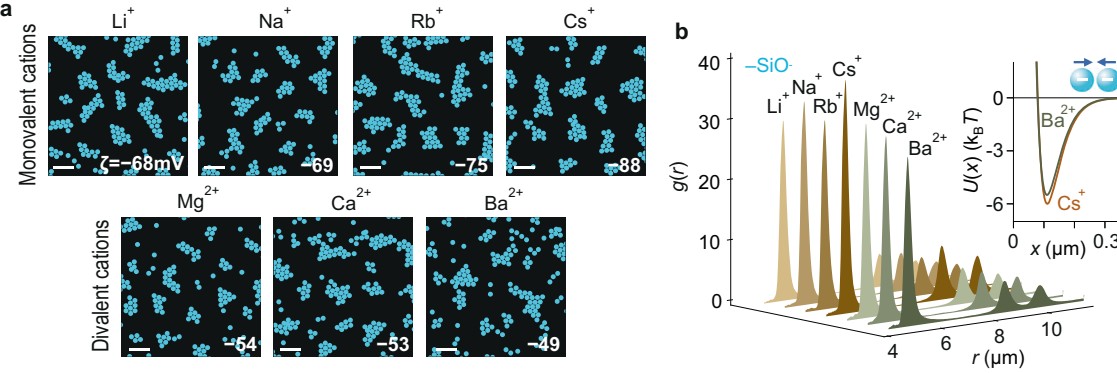

**Fig. 5 | Cluster formation in aqueous electrolytes containing monovalent and divalent cations. a** Digitised experimental image of negatively charged $SiO_2$ colloidal suspensions forming clusters in the presence of LiCl, NaCl, RbCl, CsCl, $MgCl_2$, $CaCl_2$, and $BaCl_2$ at an ionic strength of 0.1 mM. The scale bars represents 20 μm. **b** Measured $g(r)$ profiles in each case, with inferred pair interaction potentials $U(x)$

presented for $Cs^+$ and $Ba^{2+}$ (inset). Note that at the rather small interparticle separation $x \approx 0.1$ μm characteristic of these experiments, the estimated attractive contribution to the pair potential from the van der Waals force is $\approx 1\,k_B T$, which is larger than in the rest of the experiments in the study but significantly smaller than the measured minima of depth $w \approx 6\,k_B T$.

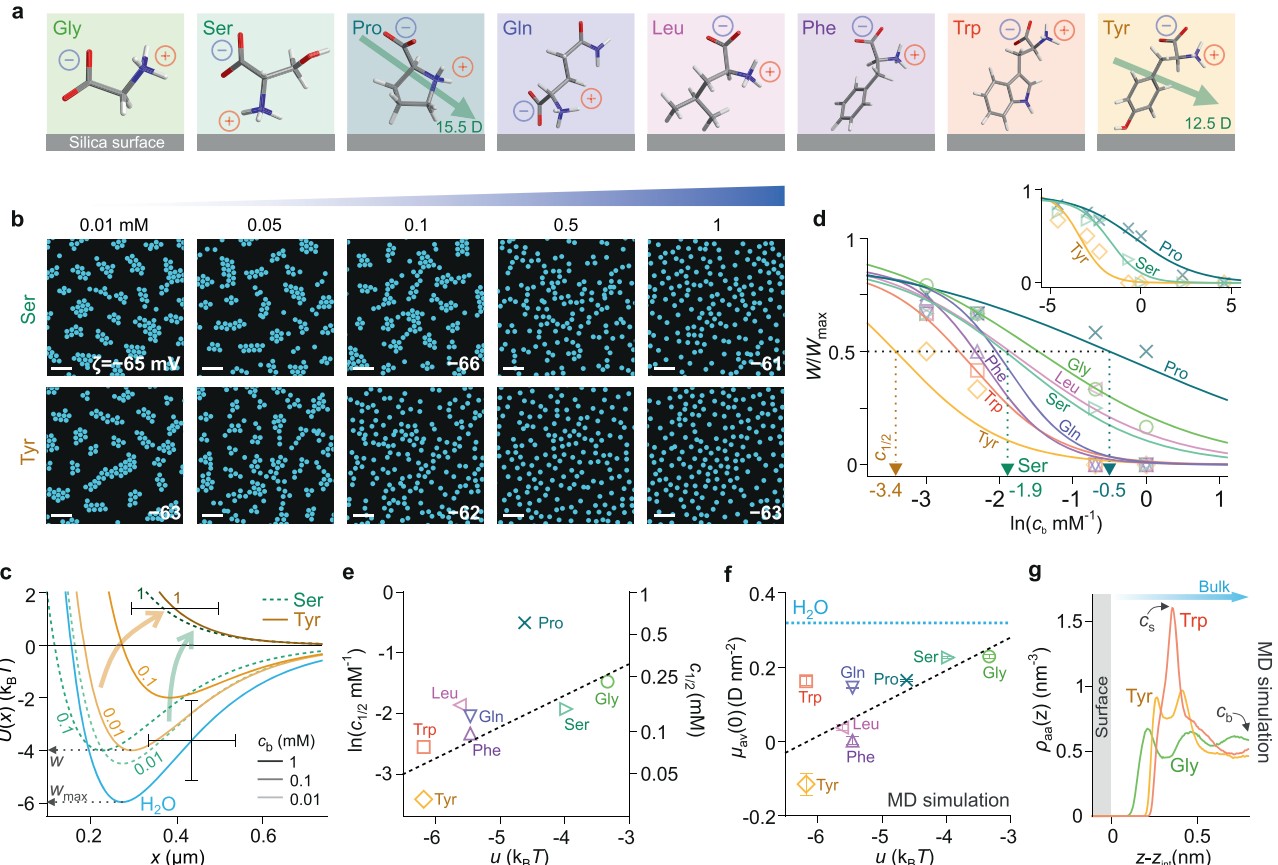

**Fig. 6 | Amino acids can abolish cluster formation at very low concentration. a** Molecular structure of amino acids investigated. Molecular orientations depict averages observed in MD simulations at a silica interface. Average interfacial dipole moment vectors for proline and tyrosine (green arrows) imply normal components that are oriented downward, opposite to that of pure water at an interface (see Fig. 2a). **b** Digitised images of negatively charged $SiO_2$ particles suspended in various concentrations of net-neutral amino acids in water, shown here for serine and tyrosine (see Supplementary Fig. 24 for all amino acids studied). The interparticle attraction significantly weakens at $c_b = 0.5$ mM for serine and $0.1$ mM for tyrosine and vanishes at higher concentrations. Scale bars 20 μm. **c** Inferred pair-interaction potentials $U(x)$ for serine and tyrosine-containing solutions inferred from BD simulations (see Supplementary Table 28 for parameters). Error bars denote estimated uncertainties of $\pm 100$ nm on particle diameter and $\pm 1.5$ $k_BT$ in $w$. **d** Normalised well depths, $w/w_{max}$, as a function of $c_b$ with sigmoidal fits to the data

(coloured lines). At $c_b = c_{1/2}$, the clustering strength is half-maximal, i.e., $w/w_{max} = 0.5$. **e** Plot of measured $\ln c_{1/2}$ vs. PMF minima values, $u$, taken from MD simulations in ref. 44 (symbols) with a linear fit (black dashed line). **f** Excess interfacial dipole moment densities, $\mu_{av}(0)$, from MD simulations of $c_b = 1$ M aqueous solution of amino acid in contact with a neutral silica surface. Error bars depict uncertainty arising from simulation convergence (see MD simulation methods). $\mu_{av}(0)$ values for the amino acid-containing electrolyte are substantially lower than the pure water value, $\mu_{av}(0) \approx 0.3$ D nm$^{-2}$ (blue dashed horizontal line), with the magnitude decreasing with the increasing surface affinity of the amino acid reflected in $|u|$. **g** Averaged amino acid density profiles in solution, $\rho_{aa}$ (as a function of distance from the silica surface ($z - z_{int}$)), for the simulations described in (**f**), present qualitative trends of the interfacial amino acid concentration, $c_s$, which is highest for tryptophan and lowest for glycine.

behaviour was observed at about an order of magnitude higher concentration ($c_b \approx 1$ mM).

Mechanical force measurements have shown some impact of molar concentrations of zwitterionic osmolytes such as glycine and trimethylglycine (TMG) on the magnitude of the electrostatic repulsion between surfaces, and that zwitterions can accumulate and layer at mica interfaces[40,41]. However, at low concentrations ($c_b < 30$ mM), these osmolytes did not influence the magnitude and screening length of the interaction between mica or silica surfaces[40,42]. Along these lines, our measured $\zeta$-potential values showed no significant change with increasing amino acid concentration in solution (Supplementary Tables 14–16). This suggests that the presence of amino acids does not measurably alter the interparticle electrostatic repulsion ($\Delta F_{el}$ term in Eq. (1)), but apparently rather dramatically impacts the long-range attraction between negatively charged particles, the likely origin of which is captured within the interfacial solvation view, as elucidated below.

We found that the concentration of amino acid at which particle clustering significantly weakens and above which it vanishes follows a

trend in interfacial adsorption affinities of the amino acid molecules to the silica surface, values of which have been obtained in previous experimental and MD simulation studies[43–45]. We estimated the magnitude of the pair potential well depths, $w$, as a function of amino acid concentration and compared these values with $w_{max}$, a measure of the electrosolvation interaction strength in pure water devoid of added solutes. In each instance, we determined the bulk concentration of amino acid, $c_{1/2}$, at which the strength of the interparticle attraction given by $w/w_{max}$ was half-maximal (Fig. 6d). Importantly, quartz crystal microbalance measurements at silica and polystyrene interfaces have shown that amino acids and polypeptides adsorb to these interfaces substantially[38,39]. The reported extents of adsorption, in general, correlate with $u$, the value of the minimum energy in the potential of mean force (PMF) for a single amino acid molecule interacting with a silica surface obtained from MD simulations in the literature[38,39,44,46]. We may therefore estimate a surface concentration of amino acid, $c_s$, for a given bulk concentration $c_b$ using the Boltzmann relation $c_s = c_b \exp\left(-\frac{u}{k_BT}\right)$[44,46]. Assuming constant $c_s$, a plot of $\ln c_{1/2}$ vs. $u$ reveals a linear relationship for the amino acids tested. This

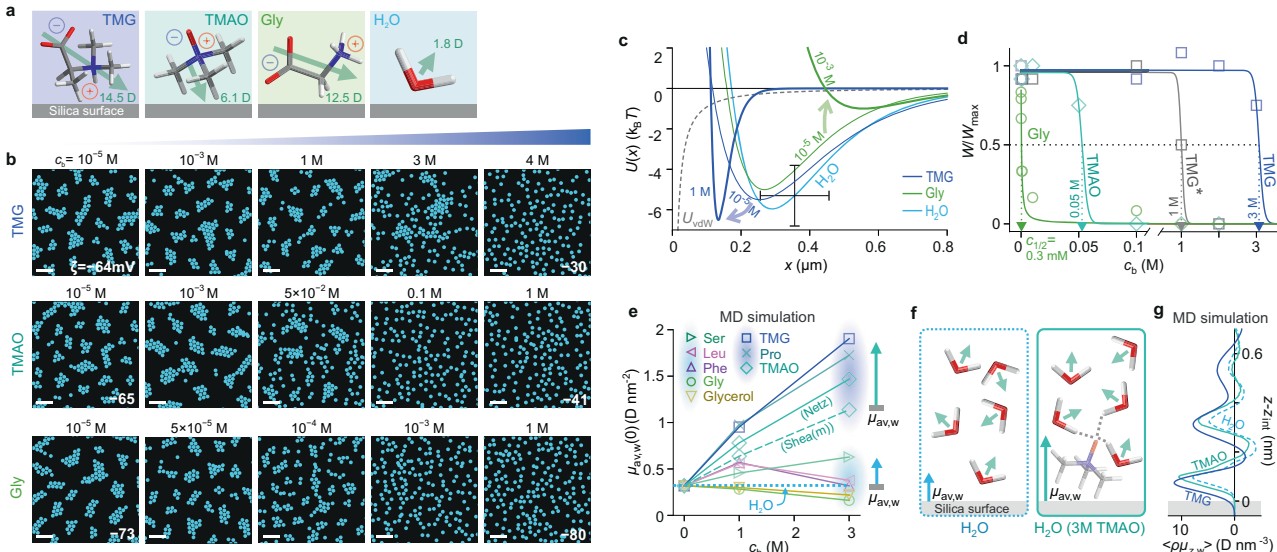

**Fig. 7 | Highly disparate influence of zwitterions on interparticle attraction.**
**a** Schematic depiction of average TMG and TMAO molecular orientation at a silica surface compared to glycine and water, inferred from MD simulations. **b** Digitised experimental images of colloidal suspension structure for negatively charged $SiO_2$ particles in aqueous solutions of increasing concentrations, $c_b$, of TMG, TMAO and glycine. Scale bars 20 μm. **c** Inferred pair-interaction potentials $U(x)$ (see Supplementary Table 29 for parameters) for $c_b = 10^{-5}$ M and higher concentrations of zwitterion. Error bars denote estimated uncertainties of $\pm 100$ nm on particle diameter and $\pm 1.5$ $k_B T$ in $w$. **d** Normalised pair-interaction potential well depths, $w/w_{max}$, as a function of zwitterion concentration in solution, indicate $c_{1/2} = 3$ M (TMG in water), 1 M (TMG solution with pH and conductivity matched to TMAO solution – TMG*), 50 mM (TMAO in water), and 0.3 mM (glycine in water). **e** Contribution of water alone to the total excess dipole moment density at the

interface, $\mu_{av,w}(0)$, in MD simulations of a silica surface with a group density of 4.7 OH nm$^{-2}$ immersed in water containing varying zwitterion concentration (see MD simulation methods). Zwitterions can be grouped into two categories depending on their qualitative effect on the sign and magnitude of $\mu_{av,w}(0)$. TMG, TMAO, and proline zwitterions enhance the value of $\mu_{av,w}(0)$ significantly above that of pure water at a silica surface (blue horizontal dashed line). Data corresponding to two TMAO models (Shea(m) and Netz) are presented (see MD simulation methods). **f** Schematic illustration of excess dipole moment density, $\mu_{av,w}$, at a silica surface in pure water, and a solution containing $c_b \approx 3$ M TMAO (see Supplementary Note 6 for detail). **g** Spatial profiles of the water dipole moment density at the interface for TMAO- and TMG-containing aqueous media (solid blue lines) are consistently more positive than the pure water case (blue dashed line), which yields $\mu_{av,w}(c_b = 3M) > \mu_{av,w}(c_b = 0)$.

suggests that the more strongly adsorbing an amino acid is (larger $|u|$), the lower the bulk concentration $c_b$ required to achieve a given surface concentration $c_s$ that perturbs the interfacial structure and induces a substantial modulation of the long-ranged attractive interaction (Fig. 6e). Indeed, the different levels of adsorption of such amino acids and the varying impact they have on the interfacial water structure has been reported in SFG experiments studying both hydrophilic silica and hydrophobic polystyrene surfaces[38,39].

Furthermore, MD simulations of water containing a fixed bulk concentration of $c_b = 1$ M amino acid in contact with a silica surface permit us to determine the total average dipole moment density, $\mu_{av}(0)$, which includes the contribution of all interfacial species – solvent and zwitterion – in each case (Fig. 6f, g). Interestingly, simulations reveal that in the presence of amino acid in solution, $\mu_{av}(0)$ decreases significantly in magnitude from its pure water value, implying a diminished contribution from the interfacial term in Eqs. (1) and (2), and a weakening of cluster formation as an experimental consequence (Fig. 6d and Supplementary Fig. 10). The similarity between the trends of both the experimentally determined $\ln c_{1/2}$ value, and simulation estimates of $\mu_{av}(0)$, as a function of amino acid surface affinity, quantified by the parameter $u$, strongly implicates altered molecular interfacial structuring in the experimentally observed long-range attraction (Fig. 6f).

We then turned our attention to cluster-formation experiments in the presence of methylated zwitterions such as trimethylglycine (TMG) and trimethylamine $N$-oxide (TMAO), which are known to play an important role in stabilising the folded, more compact structure of proteins under environmental stress, as well as in regulating the formation of phase-separated protein aggregates[11–13,15,47,48]. Infrared and terahertz spectroscopy investigations have shown that TMG and TMAO strengthen the H-bonding network and alter the dynamics of

bound hydration water, respectively, and these properties have been implicated in their ability to stabilise biomolecular folding[49,50]. We performed experiments on $SiO_2$ particles suspended in aqueous solutions at increasing concentrations of TMG in an identical manner to experiments with glycine. We found that cluster formation persisted up to a much higher concentration of TMG of $c_b \approx 3$ M compared to the amino acids, and above which the strength of the interparticle attraction diminished (Fig. 7a–d). Methylation of the amine functionality, therefore, appears sufficient to dramatically alter the impact of TMG on cluster formation and stability, rendering the $c_{1/2}$ value for TMG four orders of magnitude higher than its unmethylated counterpart. This observation is reminiscent of the progressive impact on the water structure of alkylation in osmolytes such as urea and glycine[51,52]. Methylated glycine derivatives have also been demonstrated to adsorb more strongly than glycine to silica nanoparticles[45]. However, in experiments performed in pure water, high concentrations of TMG can result in a significant increase in ionic strength in solution, due to weak protonation at pH 7 (ca. 0.01%) of the carboxyl group whose $pK \approx 3$, which in experiments on colloidal particles generally results in a weakening of the long-ranged attractive force[18]. Ionic strength measurements for 4 M TMG in solution indeed indicate $c_0 \approx 0.5$ mM which is substantially higher than $c_0 \approx 10^{-5}$ M value noted for experiments with zwitterionic amino acids (Supplementary Table 11). Thus even at the highest concentration of TMG, the reduced propensity for cluster formation may equally likely stem from an indirect slight increase in ionic strength rather than from a direct influence of the TMG molecule on the interfacial electrolyte properties[40]. Although the reasons for the stark disparity in the impacts of glycine and TMG on interparticle attraction are not immediately clear, it is possible that TMG acts to strengthen the hydrogen bonding network of water, which has been noted in both experimental

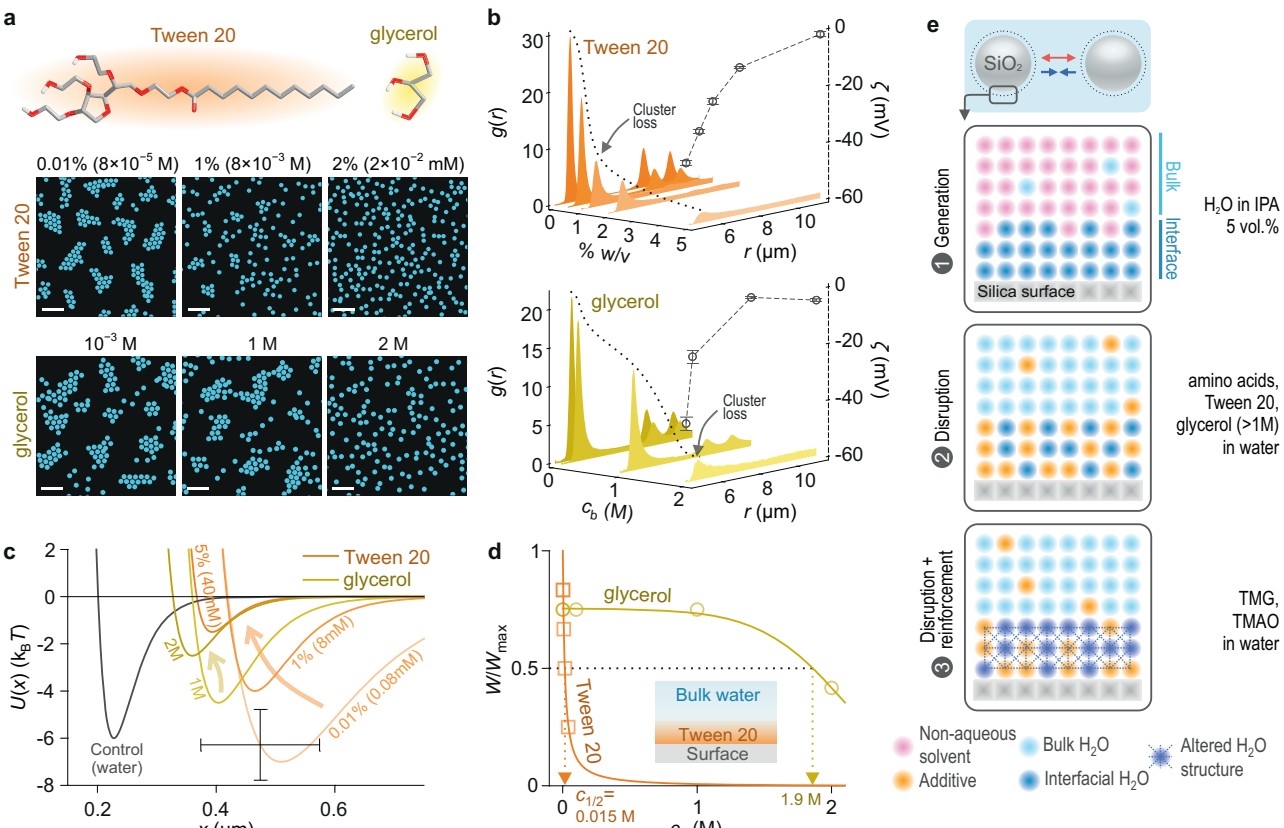

**Fig. 8 | Surfactants suppress cluster formation. a** Images of colloidal suspension structure for negatively charged $SiO_2$ particles suspended in an aqueous solution containing increasing concentrations, $c_b$, of Tween 20 and glycerol. Scale bars 20 μm. **b** $g(r)$ profiles (left axis) and measured zeta potentials ($\zeta$) (right axis) for Tween 20 and glycerol experiments. **c** Inferred pair-interaction potentials $U(x)$ as a function of Tween 20 or glycerol concentration (see Supplementary Table 30 for parameters). Control experiment in water with pH and conductivity identical to 5% Tween (grey). Error bars denote estimated uncertainties of ±100 nm on particle diameter and ±1.5 $k_B T$ in $w$. **d** Normalised pair-interaction potential well depths, $w/w_{max}$ as a function of additive concentration yield $c_{1/2} = 0.015$ M for Tween 20 and 1.9 M for glycerol. Schematic representation of possible mechanism by which Tween 20 disrupts cluster formation, depicting adsorption of surfactant to the silica surface (inset). **e** Schematic view of how microscopic interfacial structuring drives long-range interparticle forces. Electrosolvation-governed attraction can occur in solvent mixtures when, e.g., trace amounts of surface-associated water generate a thin aqueous interfacial layer (dark blue spheres – case 1). Strongly adsorbing additives (orange) such as amino acids and surfactants may disrupt interfacial water structure, thus suppressing the interparticle attraction even at low concentrations $c_b$ (case 2). Additives that potentially entail weaker disruptive effects on interfacial structure, such as glycerol, may have no significant impact on cluster formation up to high concentrations, $c_b \lesssim 1$ M. Surface-adsorbing alkylated zwitterions may reinforce interfacial water orientation (hatched region – case 3), offsetting disruptive effects due to adsorption, thereby sustaining cluster formation up to molar $c_b$ values.

and computational studies focusing on explaining the effect of TMG and TMAO on protein stability[49–51] (Fig. 8e and Supplementary Note 6).

Similar experiments on TMAO in solution yielded a $c_{1/2}$ value of 50 mM, about two orders of magnitude larger than for glycine, but nearly two orders of magnitude lower than for TMG (Fig. 7d). However, we found significant differences in the pH and ionic strength in solutions of TMAO and TMG of the same concentration. We attribute these observations to the weaker acidity of the amine oxide functionality of TMAO ($pK > 4$), resulting in a lower degree of ionisation compared to the carboxyl group of TMG and a loss of zwitterionic (neutral) character as a consequence (Supplementary Table 13). Comparing experiments that controlled for pH and ionic strength, we found the $c_{1/2}$ value for TMG decreased substantially to 1 M, closer to the value obtained for TMAO, but nonetheless about an order of magnitude larger (grey curve in Fig. 7d). The experiments thus indicate that TMAO and TMG are similar in their influence on the electrosolvation interaction and that their quantitative impact on cluster formation, as reflected in $\ln c_{1/2}$ values, is readily distinguishable from the zwitterionic amino acid family.

Importantly, $\mu_{av}(0)$ values inferred in molecular simulations for aqueous suspensions of osmolytes at a silica surface revealed no apparent differences amongst all the zwitterions examined, suggesting

similar experimental trends in cluster formation and dissolution, which was clearly not observed (Supplementary Fig. 10). However, for electrolyte containing TMG or TMAO, we found that the contribution of the water molecules alone to the excess dipole moment density at the interface, $\mu_{av,w}(0)$, displayed a large enhancement of the pre-existing molecular dipole orientation characterising pure water at a silica interface (Fig. 7e and MD simulation methods)[53,54]. At a TMG concentration of $c_b = 3$ M we noted that $\mu_{av,w}(0) \approx 2$ D nm$^{-2}$, which is approximately five times the value for pure water at the same neutral silica interface (Fig. 7e). This is in stark contrast to the other amino acids, except proline, where in general we estimated rather small changes in the value of $\mu_{av,w}(0)$ compared to pure water. Thus, as indicated by previous computational and experimental studies, our simulations suggest that TMG and TMAO have a strong ordering effect on interfacial water molecules compared to the other zwitterions considered. This property likely offsets any disruption of the interfacial electrolyte structure caused by their adsorption at the interface and may underpin their role in the cluster formation problem (Figs. 7f, g, 8e)[45,48,49,55]. Interestingly, we found that the qualitative effect on $\mu_{av,w}(0)$ of proline was similar to that of TMG at a silica interface. This not only recapitulates its higher $\ln c_{1/2}$ value compared to the other amino acids (Figs. 6e, 7e) but is also reminiscent of similarities

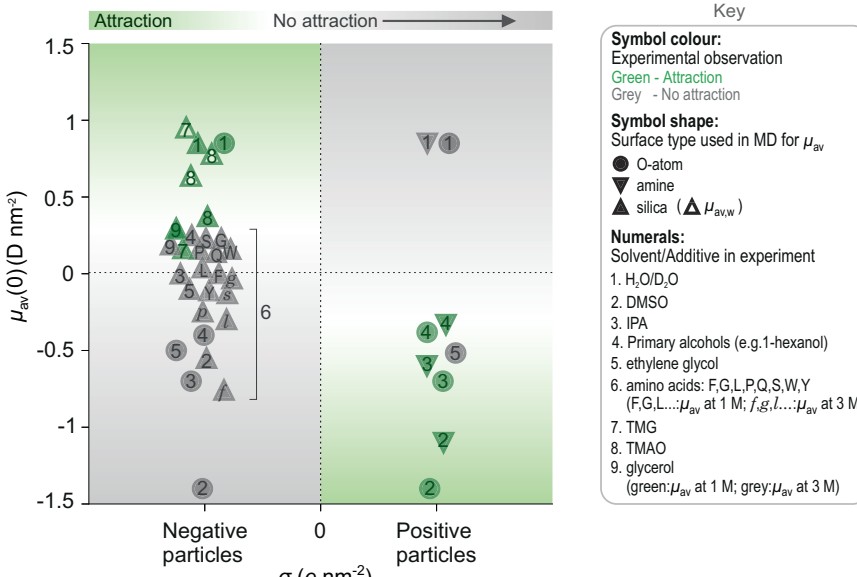

**Fig. 9 | Graphical summary and comparison of the experimental observations with expectations based on the interfacial solvation model.** Background colour on the plot presents the qualitative theoretical expectation based on the interfacial solvation model given by Eqs. (1) and (2). For negatively charged particles ($\sigma < 0$), attraction (green area) is expected when the average interfacial dipole moment density is substantial and positive, i.e., $\mu_{av} > 0$, whereas no attraction is expected when $\mu_{av} \lesssim 0$ (grey area). Conversely, for positively charged particles ($\sigma > 0$), attraction is expected when $\mu_{av} < 0$ and no attraction when $\mu_{av} \gtrsim 0$. Qualitative experimental outcomes are presented as coloured symbols on a binarized abscissa specifying the sign of the charge of the particle – positive or negative, with the corresponding ordinate value denoting values of $\mu_{av}$ (filled symbols) or $\mu_{av,w}$ (open symbols) obtained from MD simulations. Symbol colour denotes the experimental observation: attraction and cluster formation (green symbols), and absence of attraction and cluster formation (grey symbols). The graph presents results describing a total of 37 experimental situations for which $\mu_{av}$ values were determined (e.g., $\mu_{av}$ for Tween 20 and solvent mixtures at an interface are not available). All depicted dipole moment density values for silica reflect a group density of 1 OH nm$^{-2}$, except values for the amino acids, TMAG, TMAO and glycerol which entail a group density of 4.7 OH nm$^{-2}$.

amongst these three zwitterionic species observed in other contexts[42], e.g., the zwitterions TMG, TMAO and proline are considered 'protective osmolytes' for their stabilising effect on the folded state of proteins[12,14,56]. Also, SFG measurements on aqueous solutions of amino acids at silica and polystyrene interfaces show that whilst glycine does not significantly alter the signal attributed to the net orientation of interfacial water, other amino acids such as proline and leucine do change the net orientation of interfacial water considerably[39]. Furthermore, our experiments on COOH particle interactions in the presence of zwitterions in solution shed light on the role of surface chemistry in the electrosolvation interaction (Supplementary Note 9), also echoing the surface-chemistry dependence of interfacial electrolyte structure reported in spectroscopy investigations[23,39,57].

In a final experiment on the role of interfacially active agents, we examined the effect of a popular non-ionic surfactant polysorbate-20 (Tween 20), widely used in biochemical experiments to solubilise proteins and stave off non-specific adsorption to surfaces[58]. In line with its known propensity to form high-density self-organised monolayers at hydrophobic interfaces, we found that low concentrations of Tween 20 (>1% (w/v) or $c_b \approx 8$ mM) abolished stable cluster formation in silica particles (Fig. 8a–d), recapitulating the qualitative impact of this surfactant in a range of practical contexts involving molecule-surface interactions[58]. This behaviour may be contrasted with the effect of a simple triol, such as glycerol, known to stabilise folded states of proteins[59]. Glycerol had a non-disruptive influence on cluster formation up to $c_b \approx 1$ M in solution, similar to the water structure-reinforcing osmolytes tested (TMG and TMAO) (Fig. 8). Importantly, unlike for these zwitterionic osmolytes, our MD simulations suggest that glycerol present at $c_b = 1 - 3$ M in an aqueous electrolyte adsorbs to the silica interface, yet leaves both the total net dipole moment density, $\mu_{av}(0)$, and the dipole moment density due to water, $\mu_{av,w}(0)$, relatively unchanged (Supplementary Fig. 10). The implication within

the interfacial solvation model is a relatively unaltered value of $B$ in Eq. (2), and therefore the persistence of cluster formation even at relatively high concentrations of glycerol as indeed seen in experiments. A graphical summary of the experimental observations presented on a plot of $\mu_{av}$ vs. sign of $\sigma$ shows that experimentally observed trends largely align with expectations from the interfacial solvation model of interparticle interactions (Fig. 9).

Our study provides more than 40 different experimental instances where the properties of interfacial electrolyte structure – as inferred from MD simulations – correlate with the presence or absence of long-range attraction between electrically like-charged particles in a wide range of particulate systems suspended in a variety of media containing varying amounts of co-solvents and neutral solutes (Fig. 9). The experimental outcomes are summarised in a plot that uses the sign of $\sigma$ as inferred from electrokinetic (zeta potential) measurements on the particle sample and $\mu_{av}$ values from MD simulations (Fig. 9, Supplementary Fig. 2 and Supplementary Tables 3–24). We find that the presence or absence of the long-range interparticle attraction can be explained within a view that invokes either the net excess dipole moment surface density in the electrolyte or that of the solvent alone, in the immediate vicinity of the interface (Figs. 8e, 9). Overall, the findings provide an improved microscopic understanding of the origin of the experimentally observed electrosolvation force. The ability to modulate the long-range attraction using small neutral molecules and surfactants at low concentrations and the indications that emerge point to the profoundly important role this force likely plays in mediating the colloidal stability of suspensions, and possibly in biochemical interactions involving biomolecular condensation and protein folding. Considered in conjunction with the indications from MD simulations, our experimental observations on the effects of small-molecule additives on interparticle interactions suggest three general mechanistic possibilities: (1) adsorption to interfaces at low bulk

concentrations disrupts the pre-existing hydration structure around particles, thereby quelling the electrosolvation attraction and stabilising the suspension against aggregation (colloidal stability), e.g., amino acids and glycerol at >1 M concentration (Fig. 8e); (2) surface active agents can reinforce the hydration structure around particles and thereby sustain the electrosolvation attraction and concomitant cluster formation up to much larger concentrations (e.g., TMG and TMAO, see Fig. 8e). In the biomolecular context this mechanism could be related to trends involved in the stabilisation of the folded molecular state (folding stability); (3) the additive has a non-disruptive effect on interfacial electrolyte structure, and, therefore, cluster formation up persists up to high concentrations, e.g., glycerol at $\leq 1$ M concentration (Fig. 8e). Although our present theoretical view contains the key ingredients necessary to capture the experimental behaviour, the model does not currently include any long-range effects in the electrolyte, which may well contribute to the final overall picture. Moreover, heterogeneities in particle surface properties – such as $\sigma$ and $\mu_{av}$, a characteristic feature of natural systems – may also play an important role in the electrosolvation force[60–62]. Further advances in theoretical modelling of the interaction will likely benefit from renewed scrutiny of the role of electrical fields generated by polarised solvent molecules and dipoles at interfaces immersed in non-local media[63,64].

## Methods
### Experimental methods
**Experimental set-up.** A bright-field microscope was constructed in-house to observe colloidal particle suspensions. The microscope consists of a collimated 470 nm light-emitting diode (LED, M470L4, Thor-Labs), a 10× objective (Olympus UPlanSApo) and a charge-coupled device camera (CCD, DCU223M, ThorLabs), as illustrated in Fig. 1. A pitch and roll platform (AMA027, ThorLabs) was used to maintain a level plane for the glass observation cell, with planarity monitored using a spirit level (DWL-80E, Digi-Pas).

The glass cell for microscopy (20/C/G/1, Starna Scientific) was cleaned prior to sample loading. The glass surface naturally provides a negatively charged surface for negatively charged particles (SiO$_2$ and COOH-MF) in solutions. For positively charged particles (NH$_2$-SiO$_2$), the glass cell was coated with 1% poly(ethyleneimine) (PEI) polymer solution ($M_n \approx 60$ kg mol$^{-1}$, $M_w \approx 75$ k mol$^{-1}$, analytical standard, P3143, Sigma-Aldrich) to provide a positively charged surface layer. Once the particle suspension was loaded into the cell, the top surface was sealed with a polished cover slide such that the sample was airtight and bubble free.

**Data recording and processing.** Images of the colloidal suspension were recorded once all particles in solution had settled under gravity to the bottom surface of the cell. This typically required between 1 and 50 min, depending on the density and viscosity of the solvent studied. All experimental observations were repeated between 2 and 10 times. All recordings were performed using exposure times of 0.4 ms at a rate of 5 frames per second (fps) for 150 frames, with recordings performed three times at intervals of 5 min. The recorded images were then analysed using TrackNTrace-based localisation code to track particle coordinates for further analysis[18,65].

**Procedures for preparing particle suspensions.** Particles with different surface functionalities examined in this work include silica microspheres (SiO$_2$, type 1: 4.82 μm diameter, material density 2 g cm$^{-3}$, Bangs Laboratories; type 2: 5.17 μm diameter, material density 1.85 g cm$^{-3}$, microParticles GmbH), amino-functionalized silica microspheres (termed 'NH$_2$-SiO$_2$' or 'NH$_2$', 4.63 or 3.92 μm in diameter, NH$_2$ group content >30 μmol g$^{-1}$, microParticles GmbH), and carboxyl-functionalized melamine microspheres (termed 'COOH-MF' or 'COOH', 5.29 μm diameter, COOH group content $\approx$ 400 μmol g$^{-1}$,

microParticles GmbH). The corresponding particle size distributions are shown in Supplementary Fig. 1.

For experiments in aqueous systems, all particles were suspended and centrifuged multiple times in the electrolyte medium of interest until the pH and conductivity of the supernatant converged to the required values. SiO$_2$ and COOH-MF particles were suspended in 10 mM NaOH (99.99%, Thermo Fisher) for 10 min prior to re-suspension and centrifugation in the final electrolyte. Note this step is not essential for cluster formation, but enhances cluster formation likely by increasing the extent of deprotonation of ionisable surface groups. NH$_2$-SiO$_2$ particles were not treated with NaOH prior to experiments.

For experiments in organic solvents including alcohols, DMSO and EG examined in this study, we first prepared aqueous suspensions of colloidal particles which were then centrifuged and resuspended three times in high purity ethanol to minimise moisture content in the final suspension. The particles were then resuspended in the final solvent of interest.

**Characterisation of particle electrical charge.** The sign of particle charge in each sample was determined using electrokinetic measurements of zeta potentials (Zetasizer Nano Z, Malvern Panalytical). Representative zeta potential distributions for the silica and aminated silica particles in our experiments are presented in Supplementary Fig. 2.

**Measurement of solvent and electrolyte properties.** Non-aqueous solvents were stored under dry nitrogen to limit moisture absorption from ambient air (solvent purity and supplier information provided in Supplementary Table 1). Concentrated solutions of HCl and NaOH in anhydrous 2-propanol (AH-IPA) were used to adjust pH in non-aqueous media to minimise the amount of water introduced into the medium. For all experiments in non-aqueous media, we estimate a final percentage of less than 0.004% (v/v) of water arising from the addition of acid and base. Similarly, to vary the ionic strength in experiments performed in IPA, a saturated solution of NaCl in AH-IPA solution was prepared, and dilutions subsequently performed with AH-IPA to achieve the desired final concentration in the experiment. pH and conductivity were measured for each case (pH sensor: UltraMicroISM; conductivity sensor: InLab741 ISM; SevenDirect SD23, Mettler Toledo). Failure to properly dry the solvents prior to experiments promotes cluster formation in negatively charged particles in alcohols, as reflected in the water-IPA mixture experiments in the main text. The hygroscopic nature of the solvents considered, combined with the strong propensity of water to adsorb to the hydrophilic silica surface, likely generates an interfacial layer of water surrounding the particles, which can then dominate the observed interactions.

The ionic strength, $c_0$, of the various electrolyte solutions in our experiments were inferred by measurements of electrical conductivity, $s$, using the conductivity metre. An experimentally determined calibration curve of standard solutions with slope $a = \frac{s}{c_0}$, was used for this purpose, where $a \approx 150$ μScm$^{-1}$mM$^{-1}$. For water, we may compare this value with a theoretical estimate of the slope, $a = \frac{e^2 N_A}{6\pi\eta a_h}$ where $\eta = 0.89$ cP is the viscosity of water at 298 K, $a_h$ is the average hydrodynamic radius of the ions in solution, $e$ the elementary charge and N$_A$ the Avogadro number[66]. Using ionic radii 1.01 and 1.82 Å for Na$^+$ and Cl$^-$ ions given in ref. 67, this expression yields a value of $a = 65$ μS cm$^{-1}$mM$^{-1}$ which is within about a factor 2 of our calibration result. To convert the measured electrical conductivity to salt concentration in non-aqueous solutions such as longer linear alcohols, DMSO, and EG, in which inorganic salts may be poorly soluble, we used the same calibration relationship as for aqueous electrolytes, but corrected the inferred concentrations for the viscosity of the solvent as in ref. 18. This procedure assumes that the hydrodynamic radii for the ionic species are identical in both water and alcohol. The uncertainties involved in

the conversion of electrical conductivity measurements to salt concentration in organic solvents thus place a limit on the accuracy of experimental estimates of ionic strengths.

For experiments containing low concentrations of added osmolytes and surfactants, inferences of ionic strength were performed by measuring electrical conductivity and converting to $c_0$ as described above. For experiments involving high concentrations of added osmolytes, e.g., TMG and TMAO, we performed a viscosity correction of electrical conductivity data using the literature relationship, $\frac{\eta}{\eta_0} = 1 + Bc_b + Dc_b^2$, where $\frac{\eta}{\eta_0}$ is the relative viscosity of the solution containing solute at a molarity, $c_b$, and $B$ and $D$ are constants taken from Pitkänen et al.[68].

**Interparticle interactions in other non-aqueous solvents.** Although we did examine a broader range of solvents than those discussed in this study, e.g., formamide and dimethylformamide, we only present results for solvents where highly controlled experiments were possible. There are several important considerations that determined experimental feasibility and whether robust conclusions could be drawn from performing experiments in a given solvent. Firstly, controlled investigations require solvents of high purity as reflected in sufficiently low ionic strength, inferred from conductivity measurements. A typical range of electrical conductivity for strong cluster formation in aqueous solutions is ca. <120 µS cm⁻¹, whereas it tends to be much smaller, <0.5 µS cm⁻¹, for experiments in IPA. Secondly, since the appearance of the long-ranged attraction depends strongly on pH, we require solvents where the addition of small amounts of acid or base does not alter the protonation state of the solvent molecules or derivatives present in the medium. We also require solvents that are non-volatile, stable at room temperature and do not dissociate into by-products. For example, on exposure to air, formamide oxidises to give formic acid. Dimethyl formamide can contain small amounts of protonatable weak base dimethyl amine. Thus, although acetone, dimethyl formamide and formamide are H-bonding solvents and therefore of intrinsic interest alongside water and heavy water, they did not meet some of the above criteria, and experiments in these media are therefore not considered in this study[69].

## MD simulation methods

**Solvents in an 'O-atom-wall capacitor'.** To estimate the surface averaged dipole moment density, $\mu_{av}$, using molecular simulations, we used a parallel-plate capacitor system. Here, a slab of solvent is sandwiched by model solid surfaces carrying variable amounts of net electrical charge density, $\sigma$, as described in refs. 16,19. Simulation boxes of neat solvents were initialised, energy minimised and subjected to a short NVT and NPT equilibration to reach a density corresponding to 1 atm pressure. We used the SPC model for water except for simulations at a silica interface, and the CHARMM36 forcefield for all other solvents in this work, except for EG, where we employed the model of Gaur et al.[31]. The pressure was maintained with the Parrinello-Rahman pressure coupling method with only the z-dimension of the simulation box allowed to fluctuate. The equilibrated slabs of neat solvent were then inserted into the parallel-plate capacitor system. The capacitor plates entailed an area ≈ 10 × 10 nm², were separated by ≈ 4 nm in the z-direction, depending on the case simulated, and were composed of positionally restrained oxygen atoms that only support Lennard-Jones (LJ) interactions. The z-dimension of the simulation box was chosen such that any density oscillations of the solvent reached a constant bulk value in the middle of the box. A subset of randomly chosen atoms in the first layer of the left wall (in direct contact with the solvent) was assigned a positive charge (left plate) which was balanced by an equal number of randomly selected atoms assigned a negative charge on the right plate. Hydrogen bonds were constrained with the LINCS algorithm. The 3dc correction of Yeh and Berkowitz was applied to remove artificial polarisation induced by neighbouring image dipoles[70].

**Solvents and solvent mixtures in contact with a model silica surface.** The CHARMM-GUI webserver was used to generate model uncharged silica surfaces with surface group densities ranging from $\Gamma$ = 0.5 to 4.7 OH nm⁻² and were parametrised with the INTERFACE-FF forcefield description[71,72]. The silica slab was solvated on either side with a slab of the solvent of thickness ≈ 3 nm, energy minimised, and then subject to a short NVT equilibration with the v-rescale thermostat at 300 K for 50 ps. Next, an NPT equilibration of 500 ps was performed at 1 atm maintained with the Parrinello-Rahman pressure coupling method, with only the z-dimension of the simulation box allowed to fluctuate. We used the CHARMM TIP3P water model for simulations at a silica interface. For each solvent, the size of the simulation box was around 10 × 10 × 10 nm³, large enough so that any density fluctuations of the solvent could reach a constant bulk value (Supplementary Fig. 3). This initialisation procedure was followed by production MD runs in an NVT ensemble lasting 5–10 ns, with a timestep of 2 fs and with trajectory frames written every 0.2 ps. The particle mesh Ewald (PME) method was used to evaluate the long-range electrostatic interactions, using a 1 Å grid spacing and a short-range cut-off of 12 Å. The LJ interactions were smoothed over the range of 10–12 Å using the force-based switching function. Hydrogen bonds were constrained with the LINCS algorithm.

For simulations involving solutions containing osmolytes, water was described with the CHARMM TIP3P model, and parameters for the osmolyte molecules were taken from the CHARMM36 and CGENFF forcefields. For the zwitterion TMAO, however, we used the Shea(m) and Netz models in combination with the SPC/E water model for reasons explained in the following sections[53,54]. The simulation protocol was nonetheless similar to that outlined above[54]. Osmolyte-water solutions were generated such that the simulation box approximately attained the desired final osmolyte concentration after the NPT equilibration step. Production MD was performed in an NVT ensemble lasting 20 ns. We note that these simulations employed the same silica forcefield and surface group density as that used in ref. 44 - the study that determined PMF for the interaction of amino acids with the silica surface. Values of the PMF minima, $u$, from the study are presented in the plots in Fig. 6f.

We note here that estimating reliable interfacial solvent orientations and dipole moment densities from MD simulations of silica and amine surfaces immersed in binary solvent mixtures is challenging due to the sensitivity of such properties on the balance between the many types of solvent-solvent and solid-solvent interactions present in such systems. These interactions are described by the underlying forcefield parametrisations which may need careful attention to accurately model such complex systems, as noted in refs. 73,74. In particular, the behaviour of binary solvent mixtures at interfaces can be strikingly different to the bulk behaviour and further additional complexity arises from the possible competition of solvent molecules for surface sites[75]. Therefore, we do not provide estimates of $\mu_{av}(0)$ for IPA/water mixtures.

**Solvents at model amine surfaces.** Amine surfaces were modelled by creating a regular repeating arrangement of a small primary amine molecule $CH_3CH_2CH_2NH_2$ in a hexagonal close-packed configuration with group density $\Gamma$ ≈ 4 groups nm⁻² (Supplementary Fig. 3). The heavy atoms of the amine groups were positionally restrained throughout the simulations with a large force constant of 10,000 kJ mol⁻¹ nm⁻². The simulations employed the same simulation protocol as for the O-atom capacitor, with LJ walls placed at either end of the simulation cell in the z direction. Solvent molecules were sandwiched between the amine surface and the opposing LJ wall. The LJ wall applies a uniform LJ 12-6 potential corresponding to that of a CG321 atom type and functions to maintain the 2 d periodicity of the system. Parametrisation for the amine molecules was performed with the CHARMM-GUI webserver. To simulate amine surfaces with a

reduced surface group density of $\Gamma \approx 1$ group nm$^{-2}$, 75% of the amine groups had their atomic charges set to zero, whilst retaining their LJ interaction terms. Although the model amine surfaces generated may not accurately reflect the surface of the aminated silica surfaces of our positively charged nanoparticles, our aim was to capture the surface chemistry to a first approximation and to contrast the behaviour of a strongly hydrophilic silanol surface with a chemically different hydrogen-bonding (NH$_2$) group.

**Determining $\mu_{av}$ from MD simulations.** The excess net normal dipole moment surface-density, $\mu_{av}$, for each MD simulation was estimated according to Eq. (3). The calculation of this quantity is closely related to the that of the excess interfacial electrical potential, $\varphi_{int}$ described extensively in refs. 16,19., with the two quantities related as follows:

$$\frac{\mu_{av}(\sigma)}{\varepsilon_0} = \frac{1}{\varepsilon_0}\left[ \int_{z_{int}}^{z_{int}+l} \langle \rho(z)\mu_z(z) \rangle dz \Big|_{interface} - \int_z^{z+l} \langle \rho(z)\mu_z(z) \rangle dz \Big|_{bulk} \right]$$
$$= -\varphi_{int}(\sigma)$$

(3)

Here, $\mu_z(z) = \vec{\mu}(z).\vec{n}$ represents the normal component of the molecular dipole moment, at a distance $z$ from the interface, in a given simulation snapshot at a given spatial location with respect to the surface normal, $\vec{n}$, evaluated for any surface charge density, $\sigma$, of interest. $\rho(z)$ represents the density of the solvent molecules as a function of $z$, where the thickness of the interfacial layer is given by $l$, and $\varepsilon_0$ is the permittivity of free space. Furthermore, $\langle \ldots \rangle$ denotes spatiotemporal averaging in the $xy$ plane of the simulation box over the duration of the simulation. The polarization $P(z)$, or the average dipole moment density, $P(z) = \langle \rho(z)\vec{\mu}(z).\vec{n} \rangle |_{bulk}$, calculated at the midplane of the capacitor from the MD simulations, was found to agree well with the value expected for a capacitor with continuum water as the dielectric material of relative permittivity $\varepsilon$ as in refs. 16,19. In the interfacial region, however, the average dipole moment density, $\langle \rho(z)\vec{\mu}(z).\vec{n} \rangle |_{interface}$, departs substantially from the continuum value due to symmetry breaking in the orientational behaviour of the solvent induced by the presence of the interface. Integrating the quantity $\langle \rho(z)\vec{\mu}(z).\vec{n} \rangle$ over the interfacial region of thickness $l$, and subtracting the value of the same integral evaluated over a layer of the same thickness $l$ in the bulk liquid (located in the middle of the capacitor), gives the "excess" net normal dipole moment surface density, $\mu_{av}$, at any given value of surface charge density, $\sigma$. We note here that the spatial binning of the molecular dipole moments was performed using the O-atom coordinate for water molecules and the molecular centre of geometry for all other solvent molecules and osmolytes[19]. The spatial extent of the interfacial region is defined by the parameters $z_{int}$ and $l$. $z_{int}$ denotes the location of an interfacial plane, which for an O-atom wall is given by the location nearest to the surface where $\langle \rho(z)\vec{\mu}(z).\vec{n} \rangle$ drops below the value in the bulk solvent, as in ref. 16. The value of $l$ is determined by the distance at which oscillations in $\langle \rho(z)\vec{\mu}(z).\vec{n} \rangle$ decay to yield a constant 'bulk' value (Supplementary Fig. 3). For all systems other than the O-atom capacitor, we determined $\mu_{av}$ at a net uncharged surface only, i.e., $\mu_{av} = \mu_{av}(0)$, and used the sign and magnitude of this quantity to interpret experimental observations. Since the relevant charge densities in experiments are often expected to be quite small, i.e., $|\sigma| < 0.1$ $e$ nm$^{-2}$, and $\mu_{av}$ generally does not depend strongly on $\sigma$, except in some cases, e.g., DMSO, the value of $\mu_{av}(0)$ generally proves sufficient to rationalise experimental observations within the interfacial solvation model.

Since our model silica systems have two surfaces in contact with the solution of interest, $\mu_{av}(0)$ was determined as an average over both interfaces. Furthermore, for simulations involving osmolyte molecules in the solvent phase $\mu_{av}(0)$ was found to not converge to a constant value in the bulk liquid due to the low concentration of zwitterions in the simulation box. The residual uncertainty in $\mu_{av}(0)$ was estimated by

the difference in the value obtained for the two silica interfaces (containing contributions both from the solvent and osmolyte molecules), and was found to be small: $\Delta\mu_{av} \approx 0.01 - 0.03$ D nm$^{-2}$. Error bars depicted in Fig. 6f reflect this source of error. The interfacial dipole moment density was integrated over an interfacial region of thickness $l = 1.5$ nm for all simulations involving zwitterionic osmolytes.

**Simulations of EG.** For all simulations involving EG, we employed the model of Gaur et al.[31]. This model has been developed to correctly reproduce the ratio of gauche and trans conformers of EG in the bulk liquid and at the air/water interface. We performed simulations of EG in contact with a hydrophobic O-atom wall and silica surfaces of various silanol group densities ranging from $\Gamma = 0.5 - 4.7$ OH nm$^{-2}$. This yielded $\mu_{av}(0)$ values of $\approx -0.5$ D nm$^{-2}$ for the O atom wall and a value of significantly lower magnitude, $\mu_{av}(0) = -0.1$ to $+0.15$ D nm$^{-2}$, for the silica surfaces (Supplementary Note 3).

**Simulations of TMAO/water solutions.** The parametrisation in MD of certain zwitterions, such as TMAO, is known to be challenging, and it is important to use a model that accurately describes the solvation thermodynamics[54]. In this work, we have employed the Shea(m) and Netz models for TMAO, which are optimised forcefields for TMAO that better capture the density and activity coefficient of TMAO-water solutions, and are therefore generally preferred[53,54]. The SPC/E water model was used in the modelling of TMAO-water solutions as in ref. 54. Forcefield parameters for the TMAO models are compatible with our silica forcefield INTERFACE-FF, which was designed to be thermodynamically consistent with different classes of forcefields with minor adjustment[71,76]. Simulations of TMAO in water using the Shea(m) and Netz models displayed increasing positive values of $\mu_{av,w}(0)$ with increasing concentrations of TMAO in the aqueous medium (Fig. 7e). However, the CHARMM model (the forcefield employed for all other zwitterions in this work), resulted in significantly different orientational behaviour of TMAO at a silica interface, and a decrease in in $\mu_{av,w}(0)$ with increasing TMAO concentration such that $\mu_{av,w}(0) \approx 0$ D nm$^{-2}$ at $c_b = 3$ M. This highlights the general importance of accurate forcefield parametrisation in extracting experimentally relevant values of $\mu_{av,w}$. Whilst we cannot rule out that further model parametrisation of MD models may be needed for other zwitterionic osmolytes, we were able to qualitatively capture the general experimental trends for all other osmolytes in this study using the CHARMM models.

**Hydrogen bond analysis.** In our simulations, we identify a hydrogen bond based on geometric criteria given by a donor-acceptor distance of less than 3 Å and a donor–H–acceptor angle of over 150°. The MDAnalysis package was used to load and perform hydrogen bond analysis on the simulation trajectories[77]. In particular, we studied the number of hydrogen bonds formed between TMAO/TMG zwitterions and water molecules both at a silica interface and in bulk solution, as discussed further in Supplementary Note 6.

## BD simulation methods
The main text presents forms of the underlying pair-interaction potentials, $U(x)$, that have been inferred from BD simulations to match the experimentally measured radial probability distribution functions, $g(r)$. The procedures underpinning the simulations have been discussed in detail in ref. 18. and will be summarised briefly here. We performed BD simulations of a two-dimensional distribution of interacting spheres using the BROWNIAN package in the Large-scale Atomic/Molecular Massively Parallel Simulator (LAMMPS) software[78].

We use a pair potential $U'(x)$ between two interacting particles of the form:

$$U'(x) = Ae^{-\kappa_1 x} + Be^{-\kappa_2 x} + U_{vdW}$$

(4)

Here, the first term represents the overall repulsive electrostatic free energy of interaction, $\Delta F_{el}(x) = A\exp(-\kappa_1 x)$, with $A > 0$ always, and the second term, $\Delta F_{int}(x) = B\exp(-\kappa_2 x)$, denotes the free energy contribution arising from interfacial solvation[2], where $B$ may be positive, negative or zero as described previously and briefly in Supplementary Note 1[16,17]. Note that $\kappa_2 < \kappa_1 \approx \kappa$ as shown in refs. 16,17. Importantly the $\Delta F_{int}(x)$ term implies an attractive contribution to the total free energy for negatively charged particles in water[16,17]. The third term represents the vdW attraction between silica particles in solution, for which we have used the expression derived in ref. 79. similar to ref. 18.

The simulations account for the experimentally determined polydispersity in particle size (Supplementary Fig. 1) at the lowest level of approximation. Whilst $r$, the inteparticle separation accounts for the size of the individual particles, the interaction potential, $U'(x)$, remains unaffected and independent of the size of the particle-pair, which would not be accurate in practice. Using a value of the Hamaker constant $A_H = 2.4$ zJ (taken from ref. 80. and which is in agreement with other literature estimates[81]), we estimated a rather small contribution for the vdW interaction, $0 \geq U_{vdW} \gtrsim -0.4\,k_BT$, to the total interaction energy at large separations, $x \gtrsim 0.2\,\mu m$, which is the relevant intersurface separation for attractive interactions observed in a majority of experiments in this work. Therefore, the vdW interaction cannot be responsible for the deep and long-ranged minima implied by the clusters observed in the experiment. In experiments involving $D_2O$ or aqueous solutions containing high concentrations of TMG, we found average an interparticle separation in clusters of $x_{min} \approx 0.1\,\mu m$, implying $U_{vdW} \approx -0.6\,k_BT$ which is much smaller in magnitude than $|w|$ in these cases. The presented $U(x)$ profiles which are free of the $U_{vdW}$ contribution, that is:

$$U(x) = Ae^{-\kappa_1 x} + Be^{-\kappa_2 x} \tag{5}$$

The experimentally measured $g(r)$ curve provides an initial estimate of the location of the minimum in the pair potential $x_{min}$ which can be used to guide the choice of input parameters to the BD simulation. We generally take $\kappa_1^{-1} = \kappa^{-1}$ the Debye length, which is estimated from the experimentally measured salt concentration. We then use a trial value of the interaction free energy at the minimum, $U(x_{min}) = w < 0$, to obtain initial values for the parameters $A$ and $B$ as inputs for the pair-interaction potential $U(x)$, using the equations:

$$A = -\frac{w\kappa_2 \exp(\kappa_1 x_{min})}{\kappa_1 - \kappa_2}; B = \frac{w\kappa_1 \exp(\kappa_2 x_{min})}{\kappa_1 - \kappa_2} \tag{6}$$

where we have taken $\kappa_2/\kappa_1 \approx 0.95$, as suggested in ref. 17.

Particle coordinates for the BD simulations were initialised via random particle placement in a $200 \times 200\,\mu m^2$ box at the experimentally measured particle density ($\approx 0.008$ particles $\mu m^{-2}$). Periodic boundaries were applied in the $x$ and $y$ dimensions. The $z$ coordinates of the particles were fixed at a constant height throughout the simulation, ensuring a 2 d system of interacting particles mimicking experiment. Convergence of the potential energy per particle in our BD simulations was monitored over time. Particle positions used for the calculation of the final simulated radial probability density distributions, $g(r)$, were collected once the value of the potential energy reached a stationary value. This criterion was typically met after $\approx 30$–$60$ min of simulation time in a simulation involving a strongly attractive $U(x)$ with a well depth $|w|$ of several $k_BT$.

### Reporting summary

Further information on research design is available in the Nature Portfolio Reporting Summary linked to this article.

## Data availability

The data that support the findings of this study are available from Figshare https://doi.org/10.6084/m9.figshare.c.7183569 and from the corresponding author upon request.

## Code availability

The code used in this study are available from Figshare https://doi.org/10.6084/m9.figshare.c.7183569 and from the corresponding author upon request.

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

## Acknowledgements

European Research Council (ERC) under the European Union's Horizon 2020 research and innovation programme No 724180 (S.W., R.W.G., and M.K.).

## Author contributions

S.W. performed the experiments. R.W.-G. performed and analysed simulations. S.W. and R.W.-G. analysed the data and participated in manuscript preparation. B.W. and B.L. contributed to experiments. M.K. designed and supervised the study. R.W.-G. and M.K. wrote the manuscript.

## Competing interests

The authors declare no competing interests.
