## [Transparent Peer Review file · Nature Communications]

Chemical control of colloidal self-assembly driven by the electrosolvation force

Corresponding Author: Professor Madhavi Krishnan

Version 0:

Reviewer comments:

Reviewer #1

(Remarks to the Author)

This paper investigates the modulation of the attraction/repulsion of colloidal particles in a wide range of solutions. This work builds on prior studies, e.g. "Interfacial solvation can explain attraction between like-charged objects in aqueous solution" by Kubinocova, Hühnenberger and Krishnan and "A charge-dependent long-ranged force drives tailored assembly of matter in solution" (Wang, Walker-Gibbons, Watkins, Flynn Krishnan) published in Nat. Nano just this year. The first paper in this list already demonstrated an elegant explanation of the far-ranging attraction of negatively charged objects in aqueous solution for which DLVO theory would have been insufficient. In the present paper, the authors continue to postulate an "electrosolvation force", which they argue results from a slight difference in the orientation of the solvent (water) near the surface. After postulating this in eq. 1 the authors provide in Fig 1-5 quite interesting examples of how they can modulate the interaction of positively and negatively charged particles in terms of solvents, cosolvents like alcohol or amino acids, zwitterions and surfactants.

The authors observe self-assembly of like-charged colloidal particles, specifically anionic colloids in aqueous systems, without aggregation for cationic colloids or vice versa in e.g. alcohols on large micrometer length-scales. This was related to different dipolar orientations at their interface opposed to the bulk. They postulate that the aggregation behavior of the colloidal particles is governed by surface dipole moments on the surface and the presence of H-bonding propensity of the solvent. Overall, the results are interesting, nicely presented and of general interest to a wide audience. There is a really extensive SI with almost 30 figures. It is clearly a substantial amount of work.

However, a few minor points aside I have 2 major problems with the manuscript:

1) The results present are only very weakly related to the concept of the electrosolvation force, which is only discussed in the SI. In eq. 1 the authors claim that there are 2 length-scales, one resulting from the Coulomb interaction, another from some "dipole layer". The electrosolvation force seems to materialize when the force from the dipole layer overcomes the Coulomb interaction, but the paper lacks evidence for this.

2) It is unclear how the effects discussed in this paper go beyond their own Nat. Nano paper from this year.

Regarding point 1: In line 386 of the paper the authors write: "Our findings cement a microscopic understanding of the experimentally observed electrosolvation force in a wide range of particulate systems suspended in a variety of media containing varying amounts of co-solvents and neutral solutes. We find that the observations can be explained within a view that invokes either the net excess dipole moment surface density in the electrolyte, or that of the solvent alone, in the immediate vicinity of the interface, as indicated by molecular simulations (Figure 5e)."

These conclusions are unsubstantiated by the data. If this were the case, I would expect some figure/phase diagram of "excess dipole moment" vs. whatever other axis that summarizes their findings, showing that the systems that aggregate fall in one region where the "excess dipole moment" dominates. There is no such plot and I suspect that reasons for this lies in the three totally different mechanisms enumerated in the following paragraph. I would ask the authors to reconsider/rewrite this section because without any lack of understanding of the observed effects in terms of an at least semi-quantitative theory/model we are left with a set of unrelated effects that would even struggle to put under the umbrella of a single unifying "electrosolvation force". Up to line 386 the manuscript is purely descriptive in terms of measurements and simulations for certain quantities related to the measurement, but an overall understanding is lacking. This would ultimately prevent me from recommending this work for Nat. Comm, because it has been previously discussed in a paper published by the same authors this year.

Minor points:

1) It is already known that the interface affects the solvation (e.g. theories of surface tension are based on the different interaction strengths at the interface) and of course the dipole moments will reorient leading to different contributions for the

solvation based on the size and charge of the introduced particle (note: opposed to regular dissolved ions we have real surfaces, because it deals with suspensions).

2) It is not differentiated between shared solvation of multiple particles and single particle solvation. For the different radii and charge it is clear that the contributions for the formation of new interfaces, especially also with varying curvature will differ.

3) The role of the counterions is practically neglected apart from the „background“ of ionic strength. There is no information about the condensation/solvent separation especially for the aggregates.

4) Why is in fig. 3E „ u “ as the minimum energy of the PMF a good parameter especially between quite different molecules? Linearity in this parameter and the corresponding arguments for this figure is unclear!

5) The formulation with respect to bulk properties as „structure-breaking“ or „making“ is discussed in literature and the current consensus is that the terms „chaotropic and kosmotropic“ should be used to refer to salting in/out effects, the Hofmeister scale and protein stability but not to imply actual solvent „structuring“. These statements should be rephrased.

Reviewer #2

(Remarks to the Author)

In the present manuscript (NCOMMS-24-36717-T) Krishnan et al. focused on understanding the fundamentals of the purported electrosolvation force between charged colloidal systems. Negatively charged (SiO₂) and positively charged (NH₃-SiO₂) colloids have been utilized in numerous solvents (H₂O, D₂O, DMSO, alcohols, ethylene glycol, etc) and in the presence and absence of zwitterionic biologically relevant small molecules. Overall, the DLVO-based solution behavior of colloids is redefined by the presence of 2 main forces on the particles. Namely, the repulsive electrostatic interaction and newly defined interfacial free energy contribution has been discussed. The latter one is mainly introduced by the authors in this and a few previous manuscripts. By altering the solvent dipole moment, the net dipolar orientation at the solvent-colloid interface can lead to a significant interfacial free energy contribution and yield a long-range colloid-colloid interaction. Alternatively, the solvent may only generate a minimal interfacial free energy contribution and yield stable non-interacting colloids.

In this manuscript, the concept of the additional interfacial interaction is quite novel and new. Moreover, the interaction should be over few kT for a long-ranged distance to be able to affect the 2D motion of colloidal particles. Overall, I find this manuscript is quite important and can be published in Nat Comm after a major revision by including new experimental results. Since it is a new type of interaction between particles is introduced, sufficient surface characterization data must be provided for a Nat. Comm. publication. The manuscript lacks a high quality experimental surface characterization of colloids in various solvents and solutions. The list of questions and concerns can be seen below.

1) The interfacial chemistry of colloid/solvent is shown to be quite important, yet only MD simulation of this interface is demonstrated. For some samples, the zeta-potential values are also provided. Yet, as it is quite well known, it measures the potential on the slipping plane, not the surface charge. This becomes an increasingly more problematic in the absence of ions in the solvent. Therefore, the zeta potential alone does not help to provide a clear picture of the surface of the particles. Moreover, the limitation of the zeta-potential measurements should be discussed in the manuscript.

There are several experimental methods that can provide surface selective information about solid-liquid interfaces. For example, the nonlinear optical methods (second harmonic, sum frequency generation) can provide information about the orientational order (that include the dipolar orientation) of solvent molecules at the interface.

Surface force apparatus measurements are another technique that enables a one-to-one comparison to the simulated/calculated data of $U(x)$ over distance provided in the manuscript. Experimental confirmation of the electrosolvation force is absolutely needed for this manuscript.

At least one but potentially a set of these measurements should be performed and included to the manuscript to demonstrate the surface component of the electrosolvation force to the audience without any doubt.

2) As it is mentioned in the last page of the manuscript, the long-range interaction is necessary to assemble colloidal particles, yet the simulations cannot capture any long-range nature of the interaction. If the data does not suggest so, it could be better not to discuss this explicitly.

3) The surface of any colloidal particle is quite diverse especially glass surface. It can be seen by its wide-range PKa values (2-11) The inhomogeneities of the surface coverage and its potential implications should be discussed in the manuscript in order not to misguide general readership.

Reviewer #3

(Remarks to the Author)

Professor Krishnan and colleagues are on a roll, with several recent pioneering papers on an electrosolvation force.

The conventional assumption that like charged particles in solution always repel is clearly false, as demonstrated in earlier work by these authors. Corresponding findings have also been observed in other systems. For example, according to the conventional view, parallel packing between highly charged RNA helices is expected to be strongly disfavored by charge repulsion. Contrary to this expectation, parallel packing is observed in more than 50% of all packing interactions in crystals of A-form RNA. Clearly, the conventional view is inadequate.

Here, extending their previous work, Krishnan et al. begin to dissect the electrosolvation force by measuring the effect of

various solvents and co-solvents and by using MD simulations. These results are compared to their model (equations 1 and 2), that gives the interparticle potential as the sum of two terms: a conventional electrostatic term and an interfacial term.

The experimental protocol utilizes a well-chosen panel of solvents and relevant co-solvents, such as compatible osmolytes. Results from these experiments together with a model-based interpretation of the data provide insights and also raise some open questions.

The work presented here is detailed and well-described. A pleasure to read. -George Rose

Version 1:

Reviewer comments:

Reviewer #3

(Remarks to the Author)

The authors have addressed the reviewers' concerns satisfactorily, even expansively.

Responses to the Reviewers

Reviewer #1 (Remarks to the Author):

“This paper investigates the modulation of the attraction/repulsion of colloidal particles in a wide range of solutions. This work builds on prior studies, e.g. "Interfacial solvation can explain attraction between like-charged objects in aqueous solution" by Kubinocova, Hühnenberger and Krishnan and "A charge-dependent long-ranged force drives tailored assembly of matter in solution" (Wang, Walker-Gibbons, Watkins, Flynn Krishnan) published in Nat. Nano just this year. The first paper in this list already demonstrated an elegant explanation of the far-ranging attraction of negatively charged objects in aqueous solution for which DLVO theory would have been insufficient. In the present paper, the authors continue to postulate an "electrosolvation force", which they argue results from a slight difference in the orientation of the solvent (water) near the surface. After postulating this in eq. 1 the authors provide in Fig 1-5 quite interesting examples of how they can modulate the interaction of positively and negatively charged particles in terms of solvents, cosolvents like alcohol or amino acids, zwitterions and surfactants.

The authors observe self-assembly of like-charged colloidal particles, specifically anionic colloids in aqueous systems, without aggregation for cationic colloids or vice versa in e.g. alcohols on large micrometer length-scales. This was related to different dipolar orientations at their interface opposed to the bulk. They postulate that the aggregation behavior of the colloidal particles is governed by surface dipole moments on the surface and the presence of H-bonding propensity of the solvent. Overall, the results are interesting, nicely presented and of general interest to a wide audience. There is a really extensive SI with almost 30 figures. It is clearly a substantial amount of work.”

We thank the reviewer for their careful appraisal of our study.

“However, a few minor points aside I have 2 major problems with the manuscript:

1) The results present are only very weakly related to the concept of the electrosolvation force, which is only discussed in the SI. In eq. 1 the authors claim that there are 2 length-scales, one resulting from the Coulomb interaction, another from some “dipole layer”. The electrosolvation force seems to materialize when the force from the dipole layer overcomes the Coulomb interaction, but the paper lacks evidence for this.”

We very much appreciate this feedback and have undertaken a significant restructuring of the manuscript in response to the comments above. Because the model has been previously described and extensively experimentally tested, we have summarised the detail on the theoretical side in the SI, and retained only the two main required equations (Eqs. 1 and 2) in the main text. We focus the discussion in the main manuscript on systematically examining experimental outcomes under various experimental conditions. We consider these observations in the light of MD simulations that provide insight into the interfacial electrolyte structure. Our approach to testing the underpinning mechanism (the ability of the interfacial dipole contribution to induce an attraction) relies on carrying out a series of perturbations to the interfacial electrolyte structure and comparing the experimental consequences with expectations based on a model of the electrosolvation force (that relies on quantitative input from MD simulations). The study thus subjects the basic proposed idea behind the electrosolvation force to rigorous “chemically multidimensional” testing.

Immersing the particle in various solvents, including small amounts of a different solvent, or adding neutral molecules such as osmolytes and surfactants to the solution all tend to interfere with electrosolvation attraction. These molecules are however also generally known to alter or disrupt the interfacial electrolyte structure, while leaving the rest of the system largely unperturbed – especially at the low bulk concentrations of additives used in most experiments. The unexpected dramatic impact of these seemingly minor changes to the bulk electrolyte aligns remarkably well with the indications of a model that suggests that the observed force has its origins in the interfacial electrolyte structure which is known to be strongly influenced by surface adsorption of the species added to the solution.

Our study aims at relating the experimentally inferred sign of B , the prefactor in the electrosolvation interaction term, with the sign expected from the interfacial solvation model which requires the average interfacial dipole moment density, μ_{av} , as an input. When $\mu_{av} > 0$, we have $B < 0$ and the model would suggest an attraction is possible between negatively charged particles ($\sigma < 0$). When $\mu_{av} \lesssim 0$ however, no attraction is expected, which means that the particles are expected to display the canonical mutual repulsion evident in the lack of experimentally observed cluster formation. The opposite trend holds for positively charged particles: flipping of the signs of both μ_{av} and σ is expected to lead to attraction. This is a hallmark of what we propose as the electrosolvation interaction.

As is frequently the case of testing of any theory against experiment, our approach entails perturbing the system in a particular way, comparing the experimentally observed response with what the model would suggest, and treating the result of this comparison as evidence in favour of or against the model. We therefore performed experiments that are expected to directly affect the interfacial electrolyte structure by molecules adsorbing to the interface and altering the interfacial solvent dipole moment density as reported both in spectroscopy experiments and in MD simulations. We are able to make clear links between the persistence or dissolution of clusters, under various conditions, to the sign of interfacial dipole moment expected from simulations, strongly supporting the proposed theoretical view of an underpinning electrosolvation force.

We have significantly revised the narrative in the introduction to address this important query. We have modified Fig. 1c substantially to emphasize the conceptual basis of the study at the very outset. We have also included an additional figure, Extended Data Figure 4, to summarise the outcomes, which shows that the experimental indications align rather well with indications from the model. Whilst there is currently no clear route to quantitative experimental measurements of interfacial electrolyte structure under the variety of experimental conditions probed (as discussed further later) we draw on quantitative indications available from MD simulations, and further relate our findings to qualitative information on interfacial electrolyte structure available in the literature using spectroscopy techniques.

2) It is unclear how the effects discussed in this paper go beyond their own Nat. Nano paper from this year.

We thank the reviewer for raising in this point. In fact this work is a true sequel to the paper reporting the original set of observations. We show for the first time how we may tamper with,

modulate or perturb the long range attraction interaction based on the understanding that the mechanism behind the force is rooted in the interfacial solvent structure (which is parametrized in the theory by the net excess dipole moment density). If perturbing some part of the system elicits a response that is in line with the predictions of the postulated model or mechanism, then the observations could be taken to constitute evidence in favour of the proposed mechanism. In the Nature Nano 2024 paper we showed how pH and salt concentration in water could modulate the attractive interaction in keeping with expectations based on the model. In this study, we have greatly expanded the scope of the investigation to other solvents which illustrates how general and broadly relevant this phenomenon truly is (we show that another 10 commonly used solvents display behaviour that fall within the model). Further, in this paper we show the ability to tune the interaction by mechanisms that are expected to specifically impact the interfacial structure. For example ppm concentrations of amino acids (neutral zwitterions) in the bulk ought to leave the interaction between charged particles in solution entirely unperturbed. However due to their propensity to adsorb at an interface we may expect the interfacial electrolyte structure to be altered (as confirmed in our MD simulations, and in spectroscopy studies). Indeed we find that small amounts of surface-adsorbing zwitterions are able to entirely suppress the attractive electro-solvation interaction.

In summary, this new study provides about *42 different experimental instances* where knowledge on the interfacial electrolyte structure is examined for and related to its impact on the long range attraction. We find that the experimental observations can be largely fully rationalized within the broad expectations of the model by considering either the total dipole moment density μ_{av} in most cases, or that of interfacial water alone, $\mu_{av,w}$, in some. This suite of experiments puts to the test a core proposed feature of the electro-solvation mechanism and thus greatly fleshes out the phenomenology behind the electro-solvation force.

Given the importance of surfactants and osmolytes in molecular stability and folding, and more broadly in chemical and biochemical formulations, and the fact that relatively little is known about their mechanisms of action, this study makes an important step forward illustrating possible mechanistic implications of the electro-solvation force in a range of practical contexts. Taken together we believe this new body of evidence is a significant development in cementing the original discovery, and shedding light on the broad and general relevance of its implications.

Regarding point 1: In line 386 of the paper the authors write: "Our findings cement a microscopic understanding of the experimentally observed electro-solvation force in a wide range of particulate systems suspended in a variety of media containing varying amounts of co-solvents and neutral solutes. We find that the observations can be explained within a view that invokes either the net excess dipole moment surface density in the electrolyte, or that of the solvent alone, in the immediate vicinity of the interface, as indicated by molecular simulations (Figure 5e)." These conclusions are unsubstantiated by the data. If this were the case, I would expect some figure/phase diagram of "excess dipole moment" vs. whatever other axis that summarizes their findings, showing that the systems that aggregate fall in one region where the "excess dipole moment" dominates. There is no such plot

We thank the reviewer for raising this point. In response to the reviewer's terrific suggestion to construct a "phase diagram" summarising the observed results in relation to the interfacial

dipole moment density, we have added Extended Data Figure 4 summarising the main conclusions of the entire investigation.

We believe the original manuscript indeed failed to make the required case. The plan was that the original Fig. 1c and Table 1d, together with Eqs. (1) and (2), would lead directly to the conclusions illustrated by a plot of the kind the reviewer is referring to. Clearly the explicit presentation of such a plot is critical to crystallising the conclusions of the study and we are happy to have had the opportunity to be able to address this issue.

We have also edited Fig. 1 c to indicate at the very outset the parts of the parameter space where cluster formation is expected. Briefly, when $\mu_{av} > 0$ which is true for H₂O and D₂O in contact with a (neutral) surface, then negative particles ($\sigma < 0$) may attract (as shown in Fig. 1c). When $\mu_{av} < 0$ which is expected for a good number of solvents such as alcohols and DMSO, positive particles attract. In fact we see this as a key signature of the electro-solvation force and there are no other theoretical paradigms to our knowledge that capture both (1) the asymmetry of the attractive force to inversion of the sign of charge, as well as (2) the flipping of the effect when the estimated value of μ_{av} is expected to undergo a sign inversion. Based on this conceptual foundation, the manuscript then goes on to show how in various experiments involving particles suspended in water – where interfacially active species may be expected to alter the interfacial dipole moment density – the attractive force vanishes in a predictable way based on simply considering what Eq. (1) would predict in conjunction with inferred values of μ_{av} from MD simulations. Note that the MD simulations rely on model parameters (“forcefields”) that have been arrived at and vetted by comparison with a completely different set of experimental techniques. Absent direct measurements of interfacial dipole moment densities at particle surfaces, which are very challenging to perform (see response to Reviewer#2), we use inferences of interfacial solvent behaviour from MD simulations to build our case. We find, e.g.:

1. In alcohols – where negative particles do not attract – a small amount of water added to the bulk medium can cause attraction, which may be attributed to the build of a thin interfacial water layer with properties that may be assumed to be similar to pure water (we refer to this as “*generation*” of an interfacial dipole moment density in Fig. 5e).
2. Addition of amino acids (and surfactants such as Tween 20), which are known to adsorb to silica surfaces, are experimentally observed to snuff out the attraction. MD simulations show that amino acids adsorb to interfaces and the magnitude of μ_{av} gets significantly smaller as a result as shown in Fig. 3f and g. We term this mechanism “*disruption*” of the interfacial electrolyte in Fig. 5e.
3. Interestingly, we find that some other osmolytes (TMAO and TMG) that are known to adsorb to surfaces do not mitigate the attraction, even at high concentrations in experiment. But here we find that MD simulations show that the interfacial water dipole moment density, $\mu_{av,w}$, is *strongly reinforced* by these osmolytes (Fig. 4e and f), and thus once again we may find agreement with the prediction of Eq. (1) and an interfacial dipole moment density value suggested by MD simulations. Figs 1-4 thus all show how the μ_{av} values from simulation qualitatively and semi-quantitatively correlates with the experimental outcome indicated by Eq. (2), in line with the key “predictive element” of the electro-solvation model.
4. Other surface active agents such as glycerol at low concentrations (<1 M) hardly perturb interfacial water structure (and also entails a very small contribution to μ_{av} of its own)

up to large concentrations of 1 M. At higher concentrations of glycerol, μ_{av} indeed dips according to simulations, (since more of it accumulates at the interface, as intuitively expected), and the experimental observation is indeed that cluster formation is disrupted.

and I suspect that reasons for this lies in the three totally different mechanisms enumerated in the following paragraph. I would ask the authors to reconsider/rewrite this section because without any lack of understanding of the observed effects in terms of an at least semi-quantitative theory/model we are left with a set of unrelated effects that would even struggle to put under the umbrella of a single unifying “electrosolvation force”. Up to line 386 the manuscript is purely descriptive in terms of measurements and simulations for certain quantities related to the measurement, but an overall understanding is lacking. This would ultimately prevent me from recommending this work for Nat. Comm, because it has been previously discussed in a paper published by the same authors this year.

In fact the four suggested broad mechanisms above arose from integrating all our observed experimental outcomes (about 42 different instances) and interfacial dipole moment densities expected from MD simulations within our proposed theoretical view of the electrosolvation force. The three possible mechanisms for additives in solution – and four mechanisms overall – are intended to serve as indicative models that are capable of capturing all the experimental observations, viewed through the lens of the electrosolvation interaction, where the qualitative presence of absence of an attraction is rooted in the sign and magnitude of the interfacial dipole moment density.

Our view of the study’s conclusions is therefore rather different to the assessment provided above: we believe that our proposed model of the electrosolvation force in fact provides the crucial *unifying view* within which to mechanistically understand what may well be otherwise perceived as a collection of loosely related or even unrelated experimental observations. Since our presentation of the results failed to meet the goal of clearly communicating this point, we have rewritten parts of the manuscript and added Extended Data Figure 4 to enhance the clarity of our message.

That said, we do not by any means intend to suggest these mechanisms are proven by our study, but merely that our new mechanistic view accommodates modes of action that are readily rationalizable by the interfacial solvation mechanism underpinning the electrosolvation force (where little theoretical insight previously existed). Furnishing generalised and more direct “proof” of these three suggested modes of action in all the systems considered could entail extensive further experimentation in the future using highly sophisticated experimental techniques, such as non-linear spectroscopies performed at interfaces, that may be possible to carry out in a handful of laboratories around the world with the required expertise in these techniques. This point is discussed later in the response to Reviewer#2. However, where possible, we do point to existing interfacial spectroscopy studies at interfaces e.g., on amino acids at interfaces that have already demonstrated that the structure of the interfacial electrolyte is altered by amino acids adsorption, e.g., Somorjai’s studies (York *et al* (2009) and Holinga *et al.* (2011))

We hope the revised text and added figure addressing a key point highlighted by the reviewer have adequately addressed this very important point.

Minor points:

1) It is already known that the interface affects the solvation (e.g. theories of surface tension are based on the different interaction strengths at the interface) and of course the dipole moments will reorient leading to different contributions for the solvation based on the size and charge of the introduced particle (note: opposed to regular dissolved ions we have real surfaces, because it deals with suspensions).

Yes, we fully agree: we are not drawing upon a concept in solvation that is not known already. We are in fact bringing together ideas that have long been known in different communities to understand a key problem in interparticle interactions in fluids. Specifically, we have taken into account the thermodynamic consequences of broken orientational symmetry of solvent molecules at interfaces that has long been known in physical chemistry as underpinning the “asymmetric solvation” of ions. We attempt to incorporate the implications of these well established phenomena into a description of system behaviour in the apparently rather distantly related sphere of interparticle interactions.

2) II) It is not differentiated between shared solvation of multiple particles and single particle solvation. For the different radii and charge it is clear that the contributions for the formation of new interfaces, especially also with varying curvature will differ.

In our experiments there are no new interfaces created. Unlike possibly in theory or simulation, the particles in our experiments do not actually create new cavities in the medium. The starting point of our considerations is a medium that already contains suspended particles – i.e., all interfaces and cavities corresponding to individual particles are already formed when the experiment “begins”. The additional free energy at play in our considerations therefore solely concerns a contribution that changes when particles approach each other. Also, unlike in particle aggregation which involves very short range interactions and possible alteration in the interfacial area and hydration/solvation structure, the particles in our clusters are at huge distance from each other (ca. 0.5-1 μm). So we do not anticipate any effects from creating new interfaces, and therefore our present conceptual view of the problem does not include explicit collective effects arising from multi-particle interfacial solvation contributions. Furthermore, the radius of particles probed lies in the $\sim 2 \mu\text{m}$ regime. Since the curvature of the particles is 10^6 smaller than that of the water molecule, the radius of the particle is unlikely to influence the process in any obvious way. However, one aspect of the problem where the radius of the particle does play a role concerns the magnitude of B in Eq. (2), which we expect to have an R^2 dependence as shown previously (Behjatian *et al.* Langmuir 2022). The reason behind our choice of the $R = 2.5$ micron regime for these investigations is that, apart from being easy to observe using optical microscopy, this size scale gives an interparticle interaction energy that is large enough to form stable clusters, and yet small enough to permit good spatial sampling of the potential well on reasonable timescales (important in direct measurements of the pair potentials not shown in this work, but addressed in other studies).

3) The role of the counterions is practically neglected apart from the „background“ of ionic strength. There is no information about the condensation/solvent separation especially for the aggregates.

We have added Extended Data Figure 3 to address the role of counterion type in the experiments in water. Briefly we find no significant influence of counterion species on cluster formation at the low ion concentrations that support the observation of deep minima and stable cluster formation. This is in fact in keeping with the interfacial solvation model. This is not to say that ion-specific effects are ruled out in general, but only that ion type does not appear to be important in the experimental regime corresponding to the present set of investigations (low ionic strength and large particles).

This may be a good point at which to emphasize that we generally prefer to make a technical distinction between clusters and aggregates, in that cluster formation driven by the electrosolvation force generally results in minima at long range. This makes clusters rather dynamic, and possessing considerable internal particle mobility. Aggregates on the other hand may refer to assemblies where particles are held together by rather short range attractions and are effectively in contact. Although the distinction is likely to appear largely semantic, we felt it worth emphasizing just in case the static image data presented in the manuscript has inadvertently created a perception of highly dense close packed arrangements of particles, prompting some of the queries above.

4) Why is in fig. 3E „ u “ as the minimum energy of the PMF a good parameter especially between quite different molecules? Linearity in this parameter and the corresponding arguments for this figure is unclear!

We are happy to further clarify a rather important aspect of the study. The experiments provide clear evidence that a surface-adsorbing species added to the aqueous phase mitigates and indeed abolishes the attractive electrosolvation force. If we make the simplest possible assumption that the disruption of net dipole moment due to the solvent requires a certain interfacial concentration of our surface-active species, c_s , then assuming a Boltzmann distribution of the species throughout the system given by $c_s = c_b \exp\left(-\frac{u}{k_B T}\right)$, the logarithm of the bulk concentration, c_b , of the additive at which the attraction vanishes must be linearly related to its interaction energy u with the interface. Indeed this is what is observed in the experiment. The aromatic amino acids are the strongest surface adsorbers and are characterised by larger $|u|$ values according to MD simulations and previous experimental work. In line with this, our experiments show that elimination of the attractive long range force occurs at much lower concentrations of the strong adsorbers e.g., tyrosine, tryptophan and phenylalanine compared to the weaker adsorber glycine. In simpler terms, in experiments we find that the strongest surface adsorbers interfere with the electrosolvation force at the lowest of concentrations.

It is worth pointing out that we further confirm using MD simulations of a silica interface immersed in water containing amino acids species at the same fixed bulk concentration (e.g., 1 M), that the net reduction in total interfacial dipole moment is linearly related to the adsorption energy of the amino acid. The experimental trends are therefore readily rationalized within the electrosolvation view by (1) the higher adsorption affinity of the amino acid species to interfaces and (2) magnitude of change of the total interfacial dipole moment for a fixed concentration of amino acid species in the simulation box. In other words amino acids that adsorb most strongly to the surface largely result in the greatest change in total interfacial dipole moment density as revealed by MD. Within the interfacial solvation model

this should lead to a mitigation of the electrosolvation force at bulk concentrations, $c_{1/2}$, such that $\ln c_{1/2}$ is linear in u , a quantity characterizing the magnitude of the amino acid-surface affinity. Importantly both surface force apparatus (SFA), sum frequency generation (SFG) and quartz crystal microbalance (QCM) measurements at silica and polystyrene interfaces have clearly shown that amino acids in aqueous solution adsorb at interfaces, as pointed out in the manuscript.

5) The formulation with respect to bulk properties as „structure-breaking“ or „making“ is discussed in literature and the current consensus is that the terms „chaotropic and kosmotropic“ should be used to refer to salting in/out effects, the Hofmeister scale and protein stability but not to imply actual solvent „structuring“. These statements should be rephrased.

There was one instance in the manuscript where we referred to TMG as a “structure-maker”. We have now edited this sentence to rather state the effect TMG has been reported to display in the literature which is to strengthen the H-bonding network in water.

Reviewer #2 (Remarks to the Author):

In the present manuscript (NCOMMS-24-36717-T) Krishnan et al. focused on understanding the fundamentals of the purported electrosolvation force between charged colloidal systems. Negatively charged (SiO₂) and positively charged (NH₃-SiO₂) colloids have been utilized in numerous solvents (H₂O, D₂O, DMSO, alcohols, ethylene glycol, etc) and in the presence and absence of zwitterionic biologically relevant small molecules. Overall, the DLVO-based solution behavior of colloids is redefined by the presence of 2 main forces on the particles. Namely, the repulsive electrostatic interaction and newly defined interfacial free energy contribution has been discussed. The latter one is mainly introduced by the authors in this and a few previous manuscripts. By altering the solvent dipole moment, the net dipolar orientation at the solvent-colloid interface can lead to a significant interfacial free energy contribution and yield a long-range colloid-colloid interaction. Alternatively, the solvent may only generate a minimal interfacial free energy contribution and yield stable non-interacting colloids.

In this manuscript, the concept of the additional interfacial interaction is quite novel and new. Moreover, the interaction should be over few kT for a long-ranged distance to be able to affect the 2D motion of colloidal particles. Overall, I find this manuscript is quite important and can be published in Nat Comm after a major revision by including new experimental results.

We thank the reviewer for their support of our work. We have added new sets of experimental results as discussed in the response to Reviewer #1 and below.

Since it is a new type of interaction between particles is introduced, sufficient surface characterization data must be provided for a Nat. Comm. publication. The manuscript lacks a high quality experimental surface characterization of colloids in various solvents and solutions. The list of questions and concerns can be seen below.

1) The interfacial chemistry of colloid/solvent is shown to be quite important, yet only MD simulation of this interface is demonstrated. For some samples, the zeta-potential values are also provided. Yet, as it is quite well known, it measures the potential on the slipping plane, not the surface charge. This becomes an increasingly more problematic in the absence of ions in the solvent. Therefore, the zeta

potential alone does not help to provide a clear picture of the surface of the particles. Moreover, the limitation of the zeta-potential measurements should be discussed in the manuscript.

We fully agree with the point the reviewer is making. At the moment the only interfacial characterization technique that we are able to reasonably access, without requiring high-level expertise of external collaborators, are electrokinetic measurements that yield readouts of zeta potentials. In cases where the values are not explicitly listed in the manuscript it is because the measured value did not change significantly between the two flanking cases with values mentioned in the presentation. Although we agree that zeta potentials are not expected to be highly quantitative, they have been widely confirmed to be *qualitatively accurate* especially in high dielectric constant media (e.g., Marchioro *et al.* *J. Phys. Chem. C* (2019) and several others). Indeed we have found the sign indicated by zeta potential measurements rather reliable and consistent both with theoretical expectations and with independent experimental techniques. E.g., previous work of ours that uses a thermodynamic approach - the electrostatic fluidic trap - to measure particle charge at very low salt concentrations in various solvents showed that silica particles are negatively charged even in apolar solvents (Kokot *et al.* (2016)). Furthermore the surface charge of silica particles has already been thoroughly characterized in the literature using spectroscopy techniques ranging from SFG, SHG to XPS and there have been no surprises concerning the sign of the charge of the particle (e.g., work of the groups of S. Roke, J. Gibbs, M. A. Brown etc.). Zeta potential measurements give signs of charge that are in complete alignment with these techniques. The same is true for aminated particles which come out positive, as expected, under range of conditions. In the present study, we invoke zeta potentials only to provide a *qualitative readout of the sign of particle charge*. We do not perform any quantitative modelling or draw any conclusions that hinge on knowledge of the exact value of this charge.

Indeed, except under very rare circumstances, we have found that signs inferred from zeta potential measurements are accurate (for instance, by comparing implied readouts of the sign of charge vs. pH and comparing with theoretical expectations based on surface ionisation models). We have noted just one instance out of the 42 experimental systems investigated, (and discussed in detail in SI in the context of Fig. S4), where the sign of the measured zeta potential for silica particles in a lower dielectric constant medium (isopropanol) turned positive for low pH (ca. 4), which may simply be attributed to “double protonation” of the silanol groups. However, this measured sign was in apparent conflict with the sign inferred from the interaction of the particles with the underlying glass surface with and without a positively charged surface coating. In this case we may assume that the readout from interaction-based (thermodynamic) approach rather than the electrokinetic approach is likely to represent the correct sign of charge. Although this case was not integral to any of the conclusions it was in fact highlighted and discussed in the SI on pages 21-22.

The silica-water surface has over the past couple of decades been extensively experimentally characterized by interfacial spectroscopy methods. As cited in the revised manuscript (Refs. 25-27) second-harmonic scattering and sum-frequency generation spectroscopy studies performed on the silica-water interfaces have indeed shown exactly the pH-dependent net orientation of interfacial water as invoked in the electrosolvation interaction mechanism, and which agree with trends in MD simulations (compare for example Wang *et al.* *Nature Nano.* (2024) with Myalitsin *et al.* *J. Phys. Chem. C* (2016) and Marchioro *et al.* *J. Phys. Chem. C* (2019)). Thus, *the crux of the proposed mechanism in a pure electrolyte has already found*

independent evidence in interfacial spectroscopy investigations and we have made this point clearer in the manuscript.

In general, performing interfacial spectroscopy is a highly non-trivial endeavour, entailing the involvement of a different laboratory or several labs, and initiation of an altogether separate long-term study as described further below. The experimental set-up in these investigations generally involves flat surfaces or nanoparticles in suspension. Probing the interface of sedimented microspheres in suspension will pose a new set of practical challenges. That said, although we are unable to perform interfacial spectroscopy on the systems examined in the study, the literature already provides a wealth of evidence that amino acids, e.g., do adsorb to interfaces and can alter the orientation of interfacial water at hydrophobic surfaces and silica (e.g., papers of Somorjai cited in the manuscript). In general, the *interpretation of molecular orientations in interfacial spectroscopy is a delicate task, fraught with complexities associated with assigning reference states, making inferences on the physical origins of various spectral signatures and on the amount and relative orientations of water molecules*. In many recent experiments using SFG, e.g., the contribution from the interfacial amino acids does not show up in the signal at silica interfaces despite the fact that the substantial presence of amino acids at the interface is evident in QCM experiments. The readouts in many cases can tend to be qualitative: peaks attributed to interfacial water either grow or diminish in amplitude relative to the reference state, under various experimental conditions). In the light of all the existing data from expert labs it is not clear how much new information from interfacial spectroscopy could be added to the present state of knowledge at this time.

Given the enormous experimental challenge in quantitatively probing the structure of microsphere-electrolyte interface with the required level of accuracy, we have taken a different approach: performing MD simulations, where the relevant forcefields have themselves been carefully vetted against a range of different experimental readouts. In this approach, the ability to access the properties of the interfacial electrolyte relevant to our model in a more quantitative fashion, and under various experimentally relevant conditions, has uniquely enabled a productive comparison of experimental observations with a proposed theoretical mechanism of the interaction, the results of which are summarized in Extended Data Figure 4.

There are several experimental methods that can provide surface selective information about solid-liquid interfaces. For example, the nonlinear optical methods (second harmonic, sum frequency generation) can provide information about the orientational order (that include the dipolar orientation) of solvent molecules at the interface.

In fact, we couldn't agree more. A quantitative systematic investigation of electrolyte structure at an interface could be crucial to further examination of this important problem. Indeed, to our knowledge – as cited in our own present and prior work and outlined above – non-linear spectroscopy at interfaces has provided the only independent experimental proof of the solvent orientation effects we speak of in the manuscript and capture using MD simulations. However, because the spectroscopy-based experimental set-ups and readouts are so complex and sophisticated, a study of this nature if at all feasible only be undertaken as an independent venture, under the close direction of experts in the technique and with the engagement of their respective laboratories. Whilst we do have connections to such labs, embarking on a study of the magnitude and complexity at the particle-liquid interface is not trivial, and far from routine.

The effort will need to be well resourced and is likely to take several years to complete. Moreover, since the spectroscopic signals can generally be very complex to interpret, it is not clear whether a significant time and resource investment in this direction will even yield unequivocal readouts capable of putting numbers (both signs and magnitudes) to both interfacial solvent polarization and dipolar orientations of both water *and* various adsorbing chemical species under the relevant experimental conditions, which is the true requirement. That said, plenty of qualitative results already exist in the literature, e.g., work of Somorjai *et al.*, that show that amino acids adsorb to interfaces and perturb interfacial electrolyte structure.

In the meantime, we see our work as systematically building up a strong case for the significant undertaking entailed by a rigorous, direct and quantitative experimental probe into interfacial electrolyte structure under various conditions, that may be achieved at some point in the future. Our approach has therefore focused on employing MD simulations to provide the requisite evidence to test our newly proposed theoretical view and understanding of the experimental phenomenology. The manuscript also points to existing experimental evidence from interfacial spectroscopy wherever possible. Taken together, the availability of existing spectroscopy data, our experiments, and the clear hypotheses we put forward will likely motivate interfacial spectroscopists who may want to engage with the problem and push it further using their own sophisticated characterisation techniques.

Surface force apparatus measurements are another technique that enables a one-to-one comparison to the simulated/calculated data of $U(x)$ over distance provided in the manuscript. Experimental confirmation of the electrosolvation force is absolutely needed for this manuscript.

At least one but potentially a set of these measurements should be performed and included to the manuscript to demonstrate the surface component of the electrosolvation force to the audience without any doubt.

Again, this is a very important point. We have had extensive discussions with various experts in the surface force apparatus (SFA) technique. As cited in the manuscript surface force measurements have clearly shown the build up and adsorption of osmolytes at interfaces, to which SFA is directly sensitive, like interfacial optical techniques. In general, at the level of the repulsive part of the interparticle interaction, we are in fact full alignment with what is seen in AFM/SFA: the interaction within a range of 2-5 Debye lengths is indeed repulsive (hence the formation of stable, reorganising clusters with large interparticle spacings). However there are two key issues that stand in the way of a characterisation of the long range attractive part of $U(x)$ in any mechanical force measurement technique: (1) The force per unit radius (F/R) values implied by our measurements of attractive forces in particles is 1-2 orders of magnitude smaller than the capability of SFA and AFM, (2) Mechanical stability considerations imply that the SFA is generally better suited to the measurement of repulsions rather than attractions, and indeed most experiments in the literature focus on the measurement of repulsions. An attraction that manifests in an instrument built to measure repulsion is generally seen as a “crash into contact”. A different and technically more challenging instrument-build is required in order to measure $U(x)$ for an attractive interaction, and even if that effort were made, the force sensitivity would likely be 1-2 orders of magnitude smaller than that required. Therefore, despite having extensively explored this option, we have not been able to pursue this line of investigation with mechanical force measurement approaches.

That said, it is worth pointing out that it is optical and radiation-based methods (e.g., light and x-rays), reporting on suspension structure in solution, that have consistently reported on attraction between like charged particles, long before our present set of investigations. We also have new theoretical work in the pipeline that suggests that the conditions for the electrostatic force may be particularly well met for finite sized objects with heterogeneous properties, such as particles and molecules in solution. All these factors point to the difficulty in gaining multiple orthogonal and independent modes of confirmation of what is now a substantial set of observations in the literature stretching over decades, and spanning orders of magnitude in length scale (from molecules to particles).

Overall, we feel that the wide range of evidence we provide in the manuscript and SI (about 42 different experimental conditions) entailing experiments plus molecular simulations, and spectroscopy data in the literature, together with our original study, further builds the case that the interface plays a pivotal role in the electrostatic force. In order to specifically address the interfacial aspect of the mechanism, we have examined the role of a number of interfacially active agents and shown that the experimental observations (persistence or suppression of clusters) can be well explained by considering the impact of the interfacial agent on the interfacial dipole moment density as indicated by molecular simulations as well as in spectroscopy reports. Whilst what we show thus far furnishes strong evidence of the proposed concept, we must emphasize that proofs and demonstrations beyond doubt – if at all within reach – will require the benefit of time, as is the case with any discovery that proposes a new conceptual paradigm capable of resolving experimental anomalies that have persisted for decades.

As such, this study provides the community with a robust working model and a self-consistent view of the mechanistic underpinnings of a new interaction that aligns well with general indications from MD simulations and existing experimental data from independent techniques. These findings will hopefully spur on further investigations, not only accelerating our collective understanding of this key area but also providing the required broader community impetus for independent, orthogonal and decades-long expertise in areas such as non-linear spectroscopy and other types of force measurement to be brought to bear on the problem.

“2) As it is mentioned in the last page of the manuscript, the long-range interaction is necessary to assemble colloidal particles, yet the simulations cannot capture any long-range nature of the interaction. If the data does not suggest so, it could be better not to discuss this explicitly.”

The reviewer mentions an extremely important point. Since the present study indeed does not call for a discussion on the range of the interaction we have heeded the reviewer’s valuable advice and removed discussion of this point from the manuscript.

3) The surface of any colloidal particle is quite diverse especially glass surface. It can be seen by its wide-range PKa values (2-11) The inhomogeneities of the surface coverage and its potential implications should be discussed in the manuscript in order not to misguide general readership.

Once again, this is a crucial point. In our evolving theoretical understanding of the underlying physics, it may well turn out that particle and surface heterogeneities – germane to natural systems – play an important role in the interaction. Indeed we have discussed the heterogeneity of silica surfaces extensively in previous work, and have therefore not repeated the discussion in this manuscript. We have also added Figure S2 to the Supporting Information pdf showing the breadth of the distributions of measured zeta potentials for the particle samples under various experimentally relevant conditions. Furthermore, the concluding lines of the manuscript now point to the fact that heterogeneous surface properties may indeed play a significant role in the overall effect. We plan to work this out in future steps both on the experimental and theoretical fronts. We thank the reviewer for highlighting this important point.

Reviewer #3 (Remarks to the Author):

Professor Krishnan and colleagues are on a roll, with several recent pioneering papers on an electro-solvation force.

The conventional assumption that like charged particles in solution always repel is clearly false, as demonstrated in earlier work by these authors. Corresponding findings have also been observed in other systems. For example, according to the conventional view, parallel packing between highly charged RNA helices is expected to be strongly disfavored by charge repulsion. Contrary to this expectation, parallel packing is observed in more than 50% of all packing interactions in crystals of A-form RNA. Clearly, the conventional view is inadequate.

Here, extending their previous work, Krishnan et al. begin to dissect the electro-solvation force by measuring the effect of various solvents and co-solvents and by using MD simulations. These results are compared to their model (equations 1 and 2), that gives the interparticle potential as the sum of two terms: a conventional electrostatic term and an interfacial term.

The experimental protocol utilizes a well-chosen panel of solvents and relevant co-solvents, such as compatible osmolytes. Results from these experiments together with a model-based interpretation of the data provide insights and also raise some open questions.

The work presented here is detailed and well-described. A pleasure to read.

We thank the reviewer for their strong support of the study.